# How benign is benign overfitting?

**Amartya Sanyal**
Department of Computer Science,
University of Oxford,
Oxford, UK
The Alan Turing Institute, London, UK
amartya.sanyal@cs.ox.ac.uk

**Varun Kanade**
Department of Computer Science
University of Oxford,
Oxford, UK
The Alan Turing Institute, London, UK
varunk@cs.ox.ac.uk

**Puneet K.Dokania**
Department of Engineering Science
University of Oxford, Oxford, UK
Five AI Limited
puneet@robots.ox.ac.uk

**Philip H.S. Torr**
Department of Engineering Science
University of Oxford, Oxford, UK
phst@robots.ox.ac.uk

## Abstract

We investigate two causes for adversarial vulnerability in deep neural networks: bad data and (poorly) trained models. When trained with SGD, deep neural networks essentially achieve zero training error, even in the presence of label noise, while also exhibiting good generalization on natural test data, something referred to as benign overfitting (Bartlett et al., 2020; Chatterji & Long, 2020). However, these models are vulnerable to *adversarial attacks*. We identify *label noise* as one of the causes for adversarial vulnerability, and provide theoretical and empirical evidence in support of this. Surprisingly, we find several instances of label noise in datasets such as MNIST and CIFAR, and that robustly trained models incur training error on some of these, i.e. they don't fit the noise. However, removing noisy labels alone does not suffice to achieve adversarial robustness. We conjecture that in part sub-optimal *representation learning* is also responsible for adversarial vulnerability. By means of simple theoretical setups, we show how the choice of representation can drastically affect adversarial robustness.

## 1 Introduction

Modern machine learning methods achieve a very high accuracy on wide range of tasks, e.g. in computer vision, natural language processing etc. However, especially in vision tasks, they have been shown to be highly vulnerable to small adversarial perturbations that are imperceptible to the human eye (Dalvi et al., 2004; Biggio & Roli, 2018; Goodfellow et al., 2014) . This vulnerability poses serious security concerns when these models are deployed in real-world tasks (cf. (Papernot et al., 2017; Schönherr et al., 2018; Hendrycks et al., 2019b; Li et al., 2019a)). A large body of research has been devoted to crafting defences to protect neural networks from adversarial attacks (e.g. (Goodfellow et al., 2014; Papernot et al., 2015; Tramèr et al., 2018; Madry et al., 2018; Zhang et al., 2019)). However, such defences have usually been broken by future attacks (Athalye et al., 2018; Tramer et al., 2020). This arms race between attacks and defenses suggests that to create a truly robust model would require a deeper understanding of the source of this vulnerability.

Our goal in this paper is not to propose new defenses, but to provide better answers to the question: what causes adversarial vulnerability? In doing so, we also seek to understand how existing methods designed to achieve adversarial robustness overcome some of the hurdles pointed out by our work. We identify two sources of adversarial vulnerability that, to the best of our knowledge, have not been properly studied before: a) memorization of label noise, and b) *improper* representation learning.

**Overfitting Label Noise:** Starting with the celebrated work of Zhang et al. (2016) it has been observed that neural networks trained with SGD are capable of memorizing large amounts of label noise. Recent theoretical work (e.g. (Liang & Rakhlin, 2018; Belkin et al., 2018b;a; Hastie et al., 2019;

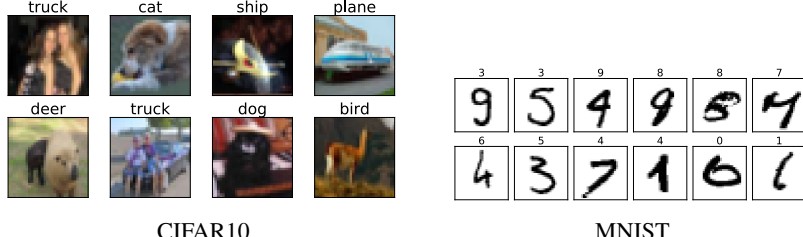

Figure 1: Label noise in CIFAR10 and MNIST. Text above the image indicates the training set label.

Belkin et al., 2019a;b; Bartlett et al., 2020; Muthukumar et al., 2020; Chatterji & Long, 2020)) has also sought to explain why fitting training data perfectly does not lead to a large drop in test accuracy, as the classical notion of overfitting might suggest. This is commonly referred to as *memorization* or *interpolation*. We show through simple theoretical models, as well as experiments on standard datasets, that there are scenarios where label noise causes significant adversarial vulnerability, even when high natural (test) accuracy can be achieved. Surprisingly, we find that label noise is not at all uncommon in datasets such as MNIST and CIFAR-10 (see Figure 1).

Our experiments show that robust training methods like Adversarial training (AT) (Madry et al., 2018) and TRADES (Zhang et al., 2019) produce models that incur training error on at least some of the noisy examples,but also on atypical examples from the classes (Zhang & Feldman, 2020). Viewed differently, robust training methods are unable to differentiate between atypical correctly labelled examples (rare dog) and a mislabelled example (cat labelled as dog) and end up not memorizing either; interestingly, the lack of memorizing these atypical examples has been pointed out as an explanation for slight drops in test accuracy, as the test set often contains similarly atypical (or even identical) examples in some cases Feldman (2019); Zhang & Feldman (2020). We point out this phenomenon for robust models through visual examples on MNIST, CIFAR10, and Imagenet (c.f. Figure 4).

**Representation Learning and Robustness:** Recent works (Tsipras et al., 2019) and Zhang et al. (2019) have argued that the trade-off between robustness and accuracy might be unavoidable. However, their setting involves a distribution that is not robustly separable by any classifier. In such a situation there is indeed a trade-off between robustness and accuracy. In this paper, we focus on settings where robust classifiers exist, which is a more realistic scenario for real-world data. At least for vision, one may well argue that "humans" are robust classifiers, and as a result we would expect that classes are well-separated at least in some representation space. In fact, Yang et al. (2020) show that classes are already well-separated in the input space. In such situations, there is no need for robustness to be at odds with accuracy. A more plausible scenario which we posit, and provide theoretical evidence in support of in Theorem 2, is that depending on the choice of representations, the trade-off may exist or can be avoided. Recent empirical work (Sanyal et al., 2020a; Mao et al., 2020) has also established that modifying the training objective to favour certain properties in the learned representations can automatically lead to improved robustness. However, we show in Section 3.2 that some training algorithms can create an apparent trade-off even though the trade-off might not necessarily be fundamental to the problem.

On a related note, it has been suggested in recent works that adversarially robust learning may require more "complex" decision boundaries, and as a result may require more data (Shah et al.; Schmidt et al., 2018; Yin et al., 2019; Nakkiran, 2019; Madry et al., 2018). However, the question of decision boundaries in neural networks is subtle as the network learns a *feature representation* as well as a decision boundary on top of it. We develop concrete theoretical examples in Theorem 2 and 3 to establish that choosing one feature representation over another may lead to *visually* more complex decision boundaries on the input space, though these are not necessarily more complex in terms of statistical learning theoretic concepts such as VC dimension.

**Summary of Theoretical Contributions**

1. We provide simple sufficient conditions on the data distribution under which any classifier that fits the training data with label noise perfectly is adversarially vulnerable.
2. There exists data distributions and training algorithms, which when trained with (some fraction of) random label noise have the following property: (i) using one representation, it is possible

to have high natural and robust test accuracies but at the cost of having training error; (ii) using another representation, it is possible to have no training error (including fitting noise) and high test accuracy, but low robust accuracy. (See Theorem 2).

The second example shows that the choice of representation matters significantly when it comes to adversarial accuracy, and that memorizing label noise directly leads to loss of robust accuracy.

**Summary of Experimental Contributions**

1. As predicted theoretically, neural nets trained to convergence with label noise have greater adversarial vulnerability. (See Section 3.1).
2. Robust training methods, such as AT and TRADES that have higher robust accuracy, avoid overfitting (some) label noise. This behaviour is also partly responsible for their decrease in natural test accuracy. (See Section 3.2).
3. To demonstrate the benefit of representation learning for adversarial robustness, we show that learning richer representation by training with more fine-grained labels, subclasses within each class, leads to higher robust accuracy. (Due to lack of space we moved this to Appendix C.3).

While our primary contribution is showing the effect of overfitting label noise on adversarial robustness, we hope our theoretical and experimental evidences on the importance of representation learning for robustness will inspire further research in this direction.

## 2 THEORETICAL SETTING

We develop a simple theoretical framework to demonstrate how overfitting, even very minimal, label noise causes significant adversarial vulnerability. We also show in Theorem 2 and 3 how the choice of representation can significantly affect robust accuracy. Although we state the results for binary classification, they can easily be generalized to multi-class problems. We formally define the notions of natural (test) error and adversarial error.

**Definition 1** (Natural and Adversarial Error). *For any distribution $\mathcal{D}$ defined over $(\mathbf{x}, y) \in \mathbb{R}^d \times \{0, 1\}$ and any binary classifier $f : \mathbb{R}^d \to \{0, 1\}$,*

- *the* natural *error is*
$$\mathcal{R}(f; \mathcal{D}) = \mathbb{P}_{(\mathbf{x},y)\sim\mathcal{D}} \left[ f(\mathbf{x}) \neq y \right], \tag{1}$$

- *if $\mathcal{B}_\gamma(\mathbf{x})$ is a ball of radius $\gamma \geq 0$ around $\mathbf{x}$ under some norm[1], the $\gamma$-adversarial error is*
$$\mathcal{R}_{\mathrm{Adv},\gamma}(f; \mathcal{D}) = \mathbb{P}_{(\mathbf{x},y)\sim\mathcal{D}} \left[ \exists \mathbf{z} \in \mathcal{B}_\gamma(\mathbf{x}) ; f(\mathbf{z}) \neq y \right], \tag{2}$$

In the rest of the section, we provide theoretical results to show the effect of overfitting label noise on the robustness of classifiers.

### 2.1 OVERFITTING LABEL NOISE

The following result provides a sufficient condition under which even a small amount of label noise causes any classifier that fits the training data perfectly to have significant adversarial error. Informally, Theorem 1 states that if the data distribution has significant probability mass in a union of (a relatively small number of, and possibly overlapping) balls, each of which has roughly the same probability mass (cf. Eq. (3)), then even a small amount of label noise renders this entire region vulnerable to adversarial attacks to classifiers that fit the training data perfectly.

**Theorem 1.** *Let $c$ be the target classifier, and let $\mathcal{D}$ be a distribution over $(\mathbf{x}, y)$, such that $y = c(\mathbf{x})$ in its support. Using the notation $\mathbb{P}_\mathcal{D}[A]$ to denote $\mathbb{P}_{(\mathbf{x},y)\sim\mathcal{D}}[\mathbf{x} \in A]$ for any measurable subset $A \subseteq \mathbb{R}^d$, suppose that there exist $c_1 \geq c_2 > 0$, $\rho > 0$, and a finite set $\zeta \subset \mathbb{R}^d$ satisfying*

$$\mathbb{P}_\mathcal{D} \left[ \bigcup_{\mathbf{s}\in\zeta} \mathcal{B}_\rho^p(\mathbf{s}) \right] \geq c_1 \quad and \quad \forall \mathbf{s} \in \zeta, \ \mathbb{P}_\mathcal{D} \left[ \mathcal{B}_\rho^p(\mathbf{s}) \right] \geq \frac{c_2}{|\zeta|} \tag{3}$$

---

[1]Throughout, we will mostly use the (most commonly used) $\ell_\infty$ norm, but the results hold for other norms.

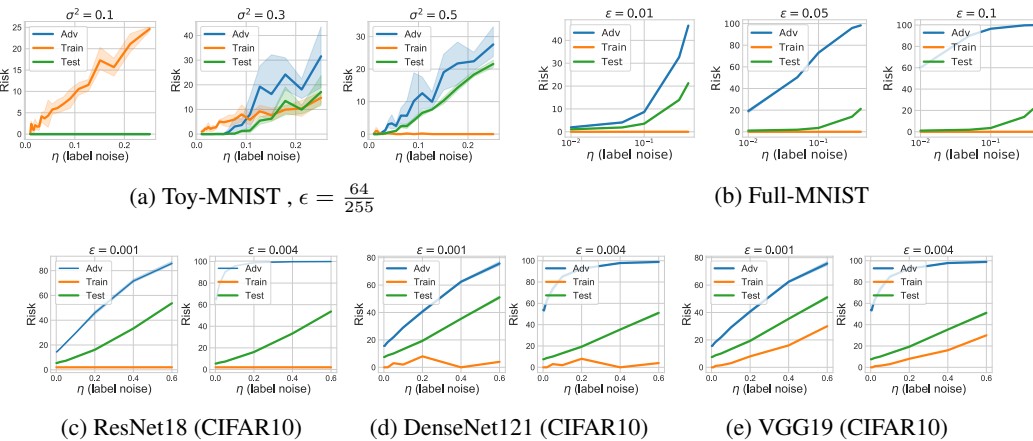

Figure 2: Adversarial Error increases with increasing label noise $\eta$. Shaded region indicates $95\%$ confidence interval. Absence of shaded region indicates that it is invisible due to low variance.

*where $\mathcal{B}_\rho^p(\mathbf{s})$ represents a $\ell_p$-ball of radius $\rho$ around $\mathbf{s}$. Further, suppose that each of these balls contain points from a single class i.e. for all $\mathbf{s} \in \zeta$, for all $\mathbf{x}, \mathbf{z} \in \mathcal{B}_\rho^p(\mathbf{s}) : c(\mathbf{x}) = c(\mathbf{z})$.*

*Let $\mathcal{S}_m$ be a dataset of $m$ i.i.d. samples drawn from $\mathcal{D}$, which subsequently has each label flipped independently with probability $\eta$. For any classifier $f$ that perfectly fits the training data $\mathcal{S}_m$ i.e. $\forall\, \mathbf{x}, y \in \mathcal{S}_m, f(\mathbf{x}) = y, \forall \delta > 0$ and $m \geq \frac{|\zeta|}{\eta c_2} \log\left(\frac{|\zeta|}{\delta}\right)$, with probability at least $1 - \delta$, $\mathcal{R}_{\mathrm{Adv},2\rho}(f; \mathcal{D}) \geq c_1$.*

The goal is to find a relatively small set $\zeta$ that satisfies the condition as this will mean that even for modest sample sizes, the trained models have significant adversarial error. We remark that it is easy to construct concrete instantiations of problems that satisfy the conditions of the theorem, e.g. each class represented by a spherical (truncated) Gaussian with radius $\rho$, with the classes being well-separated satisfies Eq. (3). The main idea of the proof is that there is sufficient probability mass for points which are within distance $2\rho$ of a training datum that was mislabelled. We note that the generality of the result, namely that *any* classifier (including neural networks) that fits the training data must be vulnerable irrespective of its structure, requires a result like Theorem 1. For instance, one could construct the classifier $h$, where $h(\mathbf{x}) = c(\mathbf{x})$, if $(\mathbf{x}, b) \notin \mathcal{S}_m$ for $b = 0, 1$, and $h(\mathbf{x}) = y$ if $(\mathbf{x}, y) \in \mathcal{S}_m$. Note that the classifier $h$ agrees with the target $c$ on *every* point of $\mathbb{R}^d$ except the mislabelled training examples, and as a result these examples are the only source of vulnerability. The complete proof is presented in Appendix B.1.

There are a few things to note about Theorem 1. First, the lower bound on adversarial error applies to any classifier $f$ that fits the training data $\mathcal{S}_m$ perfectly and is agnostic to the type of model $f$ is. Second, for a given $c_1$, there maybe multiple $\zeta$s that satisfy the bounds in (3) and the adversarial risk holds for all of them. Thus, smaller the value of $|\zeta|$ the smaller the size of the training data it needs to fit and it can be done by simpler classifiers. Third, if the distribution of the data is such that it is concentrated around some points then for a fixed $c_1, c_2$, a smaller value of $\rho$ would be required to satisfy (3) and thus a weaker adversary (smaller perturbation budget $2\rho$) can cause a much larger adversarial error.

In practice, classifiers exhibit much greater vulnerability than purely arising from the presence of memorized noisy data. Experiments in Section 3.1 show how label noise causes vulnerability in a toy MNIST model, the full MNIST and CIFAR10 for a variety of architectures.

## 3 EXPERIMENTS ON OVERFITTING LABEL NOISE

In Section 2, we provided theoretical settings to highlight how fitting label noise hurts adversarial robustness. In this section, we provide empirical evidence on synthetic data inspired by the theory and on the standard datasets: MNIST (LeCun et al., 1998), CIFAR10 (Krizhevsky & Hinton, 2009),

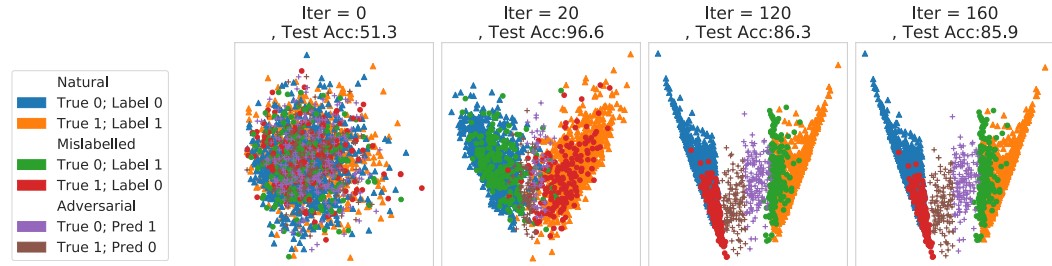

Figure 3: Two dimensional PCA projections of the original correctly labelled (blue and orange), original mis-labelled (green and red), and adversarial examples (purple and brown) at different stages of training. The correct label for *True 0* (blue), *Noisy 0* (green), *Adv 0* (purple +) are the same i.e. 0 and similar for the other class.

and on a lesser scale Imagenet to support the theory. Further details about the experimental settings including model architectures are in Appendix D.

### 3.1 OVERFITTING LABEL NOISE DECREASES ADVERSARIAL ACCURACY

We design a simple binary classification problem, *toy-MNIST*, and show that when fitting a complex classifier on a training dataset with label noise, adversarial vulnerability increases with the amount of label noise, and that this vulnerability is caused by the label noise. The problem is constructed by selecting two random images from MNIST: one "0" and one "1". Each training/test example is generated by selecting one of these images and adding i.i.d. Gaussian noise sampled from $\mathcal{N}\left(0, \sigma^2\right)$. We create a training dataset of $4000$ samples by sampling uniformly from either class. Finally, $\eta$ fraction of the training data is chosen randomly and its labels are flipped. We train a neural network with four fully connected layers followed by a softmax layer and minimize the cross-entropy loss using an SGD optimizer until the training error becomes zero. Then, we attack this network with a *strong* $\ell_\infty$ PGD adversary (Madry et al., 2018) with $\epsilon = \frac{64}{255}$ for $400$ steps with a step size of $0.01$.

In Figure 2a, we plot the adversarial error, test error and training error as the amount of label noise ($\eta$) varies, for three different values of sample variance ($\sigma^2$). For low values of $\sigma^2$ ($\sigma^2 = 0.1$), the training data from each class is all concentrated around the same point; as a result these models are unable to memorize the label noise and the training error is high. In this case, over-fitting label noise is impossible and the test error, as well as the adversarial error, is low. However, as $\sigma^2$ increases to $\sigma^2 = 0.5$, the neural network is flexible enough to use the "noise component" to extract features that allow it to memorize label noise and fit the training data perfectly. This brings the training error down to zero, while causing the test error to increase, and the adversarial error even more so. This is in line with Theorem 1.

**Remark 1.** *The case when $\sigma^2 = 0.3$ is particularly interesting; when the label noise is low and the training error is high, there is no overfitting and the test error and the adversarial error is zero. When the network starts memorizing label noise (i.e. train error gets lesser than label noise), test error still remains very low but adversarial error increases rapidly.*

We perform a similar experiment on the full MNIST dataset trained on a 4-layered Convolutional Neural Network. For varying values of $\eta$, for a uniformly randomly chosen $\eta$ fraction of the training data we assigned the class label randomly. We compute the natural test accuracy and the adversarial test accuracy for when the network is attacked with a $\ell_\infty$ bounded PGD adversary for varying perturbation budget $\epsilon$, with a step size of $0.01$ and for 20 steps and plot the results in Figure 2b. We repeat the same experiment for CIFAR10 with a DenseNet121 (Huang et al., 2017), ResNet18 (He et al., 2016), and VGG19 (Simonyan & Zisserman, 2014) to test the phenomenon across multiple state of the art architectures and plot the results in Figure 2c to 2e. The results on both datasets show that the effect of over-fitting label noise on adversarial error is even more clearly visible here; for the same PGD adversary the adversarial error jumps sharply with increasing label noise, while the growth of natural test error is much slower. This confirms the hypothesis that benign overfitting may not be so benign when it comes to adversarial error.

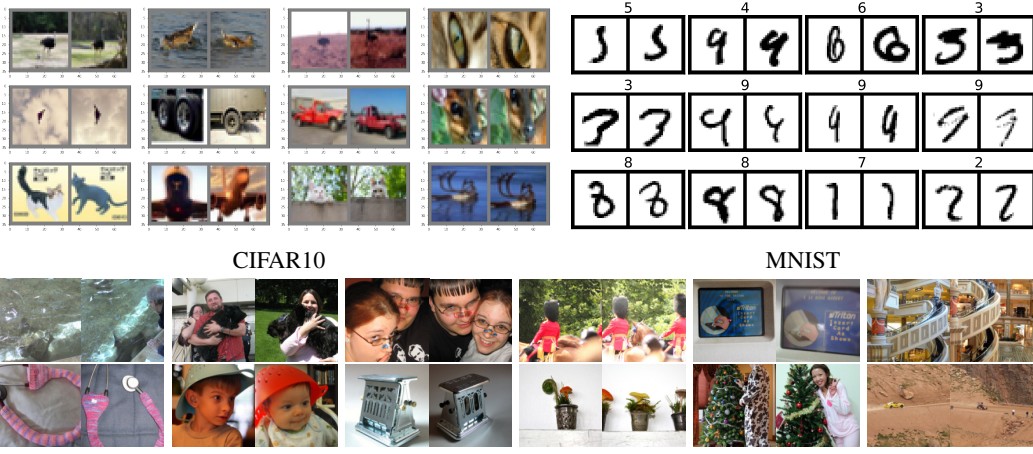

CIFAR10                                     MNIST

ImageNet

Figure 4: Each pair is a training (left) and test (right) image mis-classified by the adversarially trained model. They were both correctly classified by the naturally-trained model.

For the toy-MNIST problem, we plot a 2-d projection (using PCA) of the learned representations (activations before the last layer) at various stages of training in Figure 3. (We remark that the simplicity of the data model ensures that even a 1-d PCA projection suffices to perfectly separate the classes when there is no label noise; however, the representations learned by a neural network in the presence of noise maybe very different!) We highlight two key observations: (i) The bulk of adversarial examples ("+"-es) are concentrated around the mis-labelled training data ("○"-es) of the opposite class. For example, the purple +-es (Adversarially perturbed: True: 0, Pred:1 ) are very close to the green ○-es (Mislabelled: True:0, Pred: 1). This provides empirical validation for the hypothesis that if there is a mis-labelled data-point in the vicinity that has been fit by the model, an adversarial example is created by moving towards that data point as predicted by Theorem 1. (ii) The mis-labelled training data take longer to be fit by the classifier. For example by iteration 20, the network actually learns a fairly good representation and classification boundary that correctly fits the clean training data (but not the noisy training data). At this stage, the number of adversarial examples are much lower as compared to Iteration 160, by which point the network has completely fit the noisy training data. Thus early stopping helps in avoiding *memorizing* the label noise, and consequently also reduces adversarial vulnerability. Early stopping has indeed been used as a defence in quite a few recent papers in context of adversarial robustness (Wong et al., 2020; Hendrycks et al., 2019a), as well as learning in the presence of label-noise (Li et al., 2019b). Our work provides an explanation regarding *why* early stopping may reduce adversarial vulnerability by avoiding fitting noisy training data.

### 3.2 ROBUST TRAINING AVOIDS MEMORIZATION OF LABEL NOISE AND ATYPICAL EXAMPLES

Robust training methods like AT (Madry et al., 2018) and TRADES (Zhang et al., 2019) are commonly used techniques to increase adversarial robustness of deep neural networks. However, it has been pointed out that this comes at a cost to clean accuracy (Raghunathan et al., 2019; Tsipras et al., 2019). When trained with these methods, both the training and test accuracy (on clean data) for commonly used deep learning models drops with increasing strength of the PGD adversary used (see Table 1). In this section, we provide evidence to show that robust training avoids memorization of label noise and this also results in the drop of clean train and test accuracy.

**Robust training ignores label noise**    Figure 1 shows that label noise is not uncommon in standard datasets like MNIST and CIFAR10. In fact, upon closely monitoring the mis-classified training set examples for both AT and TRADES, we found that neither predicts correctly on the training set labels for any of the examples identified in Figure 1, all examples that have a wrong label in the training set, whereas natural training does. Thus, in line with Theorem 1, robust training methods ignore fitting noisy labels.

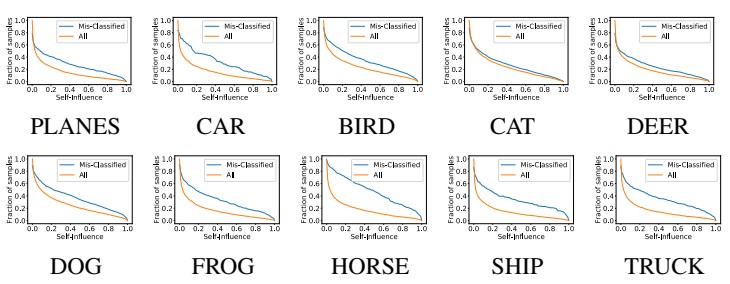 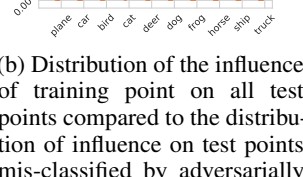

(a) Fraction of train points that have a self-influence greater than $s$ is plotted versus $s$.

(b) Distribution of the influence of training point on all test points compared to the distribution of influence on test points mis-classified by adversarially trained points.

Figure 5: The blue represents the points mis-classified by an adversarially trained model. The orange represents the distribution for all points in the dataset (of the concerned class for CIFAR10).

We also observe this in a synthetic experiment on the full MNIST dataset where we assigned random labels to 15% of the dataset. A naturally trained CNN model achieved $100\%$ accuracy on this dataset whereas an adversarially trained model (standard setting with $\epsilon = 0.3$ for 30 steps) mis-classified 997 examples in the training set after the same training regime. Out of these 997 samples, 994 examples were mislabelled in the dataset.

**Robust training ignores rare examples**   Certain examples in the training set belong to rare sub-populations (eg. a special kind of cat) and this sub-population is sufficiently distinct from the rest of the examples of that class in the training dataset (other cats in the dataset). Next, we show that though ignoring rare samples helps in adversarial robustness, it hurts the natural test accuracy. Our hypothesis is that one of the effects of robust training is to not *memorize rare examples*, which would otherwise be memorized by a naturally trained model. As Feldman (2019) points out, *if these sub-populations are very infrequent in the training dataset, they are indistinguishable from data-points with label noise with the difference being that examples from that sub-population are also present in the test-set*. Natural training by *memorizing* those rare training examples reduces the test error on the corresponding test examples. Robust training, by not memorizing these rare samples (and label noise), achieves better robustness but sacrifices the test accuracy on the test examples corresponding to those training points.

**Experiments on MNIST, CIFAR10, and ImageNet**   We visually demonstrate this effect in Figure 4 with examples from CIFAR10, MNIST, and ImageNet and then provide more statistical evidence using the notions of memorization score and influence (Zhang & Feldman, 2020) in Figure 5. Each pair of images contains a mis-classified (by robustly trained models) test image and the mis-classified training image "responsible" for it (see Appendix D.2 for details on how they were found and further details on memorization and influence.). Importantly both of these images were correctly classified by a naturally trained model. Visually, it is evident that the training images are extremely similar to the corresponding test image. Inspecting the rest of the training set, they are also very different from other images in the training set. We can thus refer to these as rare sub-populations.

The notion that certain test examples were not classified correctly due to a particular training examples not being classified correctly is measured by the *influence* a training image has on the test image (c.f. defn 3 in Zhang & Feldman (2020)). We obtained the influence of each training image on each test image for that class from Zhang & Feldman (2020) and the training images in Figure 4 has a disproportionately higher influence on the corresponding test image compared to influences of other train-test image pairs in CIFAR10. A precise notion of measuring if a sample is *rare* is through the concept of self-influence or memorization. Self-influence for a *rare example*, that is unlike other examples of that class, will be high as the rest of the dataset will *not* provide relevant information that will help the model in correctly predicting on that particular example.

In Figure 5a, we show that the self-influence of training samples that were mis-classified by adversarially trained models but correctly classified by a naturally trained model is higher compared to the

distribution of self-influence on the entire train dataset. In other words, it means that the self-influence of the training examples mis-classified by the robustly trained models is larger than the average self-influence of (all) examples belonging to that class. This supports our hypothesis that adversarial training excludes fitting these rare (or ones that need to be memorized) samples. In Figure 5b, we show that the influence of training images are higher on test images that are mis-classified by adversarially trained models as compared to an average test image from the dataset. In other words, this means that adversarially trained models mis-classify test examples that are heavily influenced by some particular training example. As we saw in Figure 5a, AT models do not memorize atypical train examples; consequently they misclassify test examples that are heavily influenced by those atypical train examples (visualized in Figure 4). This confirms our hypothesis that the loss in test accuracy of robustly trained models are due to test images that are *rare* and thus has a particularly high influence from a training image.

## 4 REPRESENTATION LEARNING AND ROBUSTNESS

Label noise by itself is not the sole cause for adversarial vulnerability especially in deep learning models trained with standard optimization procedures like SGD. A second cause is the *choice of representation* of the data, which in turn affects the shape of the decision boundary. The choice of model affects representations and introduces desirable and possibly even undesirable (cf. (Liu et al., 2018)) invariances; for example, training convolutional networks are invariant to (some) translations, while training fully connected networks are invariant to permutations of input features. This means that fully connected networks can learn even if the pixels of each training image in the training set are permuted with a fixed permutation (Zhang et al., 2016). This invariance is worrying as it means that such a network can effectively classify a tensor that is visually nothing like a real image into an image category. While CNNs don't have this particular invariance, as Liu et al. (2018) shows, CNNs are unable to predict where in the image a particular object is.

In Theorem 2, we show that for one ("correct") choice of representation, it will be impossible to fit the training data perfectly in the presence of label noise, but the classifier that best fits the training data, in that restricted class of classifiers,[2] will have good test accuracy and adversarial accuracy. On the other hand, when a different ("incorrect") representation is used, we show that it is possible to find a classifier that has no training error, has good test accuracy, but has high *adversarial error*. We posit this as an (partial) explanation of why classifiers trained on real data (with label noise, or at least atypical examples) have good test accuracy, while still being vulnerable to adversarial attacks.

**Theorem 2.** *For any $n \in \mathbb{Z}_+$, there exists a family of distributions $\mathcal{D}^n$ over $\mathbb{R} \times \{0,1\}$ and function classes $\mathcal{C}, \mathcal{H}$, such that for any $\mathcal{P}$ from $\mathcal{D}^n$, and for any $0 < \gamma < 1/4$, and $\eta \in (0, 1/2)$ if $\mathcal{S}_m = \{(\mathbf{x}_i, y_i)\}_{i=1}^m$ denotes a sample of size $m$ drawn from $\mathcal{P}$ where*

$$m = O\left(\max\left\{n \log \frac{n}{\delta}\left(\frac{(1-\eta)}{(1-2\eta)^2} + 1\right), \frac{n}{\eta\gamma^2}\log\left(\frac{n}{\gamma\delta}\right)\right\}\right)$$

*and if $\mathcal{S}_{m,\eta}$ denotes the sample where each label is flipped independently with probability $\eta$.*

*(i) the classifier $c \in \mathcal{C}$ that minimizes the training error on $\mathcal{S}_{m,\eta}$, has $\mathcal{R}(c; \mathcal{P}) = 0$ and $\mathcal{R}_{\mathrm{Adv},\gamma}(c; \mathcal{P}) = 0$ for $0 \le \gamma < 1/4$.*

*(ii) there exist $h \in \mathcal{H}$, $h$ has zero training error on $\mathcal{S}_{m,\eta}$, and $\mathcal{R}(h; \mathcal{P}) = 0$. However, for any $\gamma > 0$, and for any $h \in \mathcal{H}$ with zero training error on $\mathcal{S}_{m,\eta}$, $\mathcal{R}_{\mathrm{Adv},\gamma}(h; \mathcal{P}) \ge 0.1$.*

*Furthermore, the required $c \in \mathcal{C}$ and $h \in \mathcal{H}$ above can be computed in $O\left(\mathrm{poly}\,(n), \mathrm{poly}\left(\frac{1}{\frac{1}{2}-\eta}\right), \mathrm{poly}\left(\frac{1}{\delta}\right)\right)$ time.*

In Theorem 3, we show a similar result without the effect of label noise. In Appendix C.3, we provide experimental results learning more meaningful representations by training on more fine-grained labels (instead of the relevant coarse labels directly) can improve adversarial robustness.

---

[2]This is referred to as the Empirical Risk Minimization (ERM) in the statistical learning theory literature.

## 5 RELATED WORK

Montasser et al. (2019) established that there are concept classes with finite VC dimensions i.e. are *properly* PAC-learnable but are only *improperly* robustly PAC learnable. This implies that to learn the problem with small adversarial error, a different class of models (or representations) needs to be used whereas for small natural test risk, the original model class (or representation) can be used. Recent empirical works have also shown evidence towards this (eg. (Sanyal et al., 2020a)). In particular, the examples from Montasser et al. (2019) that uses improper learning to learn a robust classifier has a much higher sample complexity. In our example, learning algorithms for both the hypotheses classes that we use have polynomial sample complexity. Another point of distinction is that Theorem 2 uses a training set induced with random classification noise and hypothesis class $\mathcal{H}$ obtains zero training on this noisy training set whereas the examples in Montasser et al. (2019) do not have any label noise.

Hanin & Rolnick (2019) have shown that though the number of possible linear regions that can be created by a deep ReLU network is exponential in depth, in practice for networks trained with SGD this tends to grow only linearly thus creating much simpler decision boundaries than is possible due to sheer expresssivity of deep networks. Experiments on the data models from our theoretical settings indeed show that adversarial training indeed produces more "complex" decision boundaries

Jacobsen et al. (2019) have discussed that excessive invariance in neural networks might increase adversarial error. However, their argument is that excessive invariance can allow sufficient changes in the semantically important features without changing the network's prediction. They describe this as Invariance based adversarial examples as opposed to perturbation based adversarial examples. We show that excessive (incorrect) invariance might also result in perturbation based adversarial examples.

Another contemporary work (Geirhos et al.) discusses a phenomenon they refer to as *Shortcut Learning* where deep learning models perform very well on standard tasks like reducing classification error but fail to perform in more difficult real world situations. We discuss this in the context of models that have small test error but large adversarial error and provide and theoretical and empirical to discuss why one of the reasons for this is sub-optimal representation learning.

## 6 CONCLUSION

Recent research has largely shone a positive light on interpolation (zero training error) by highly over-parameterized models even in the presence of label noise. While overfitting noisy data may not harm generalisation, we have shown that this can be severely detrimental to robustness. This raises a new security threat where label noise can be inserted into datasets to make the models learnt from them vulnerable to adversarial attacks without hurting their test accuracy. As a result, further research into learning without memorization is ever more important (Sanyal et al., 2020b; Shen & Sanghavi, 2019). Further, we underscore the importance of proper representation learning in regards to adversarial robustness. Representations learnt by deep networks often encode a lot of different invariances, e.g., location, permutation, rotation, etc. While some of them are useful for the particular task at hand, we highlight that certain invariances can increase adversarial vulnerability. Thus we believe that making significant progress towards training robust models with good test error requires us to rethink representation learning and closely examine the data on which we are training these models.

## 7 ACKNOWLEDGEMENT

We thank Vitaly Feldman and Chiyuan Zhang for providing us with data that helped to significantly speed up some parts of this work. We also thank Nicholas Lord for feedback on the draft. AS acknowledges support from The Alan Turing Institute under the Turing Doctoral Studentship grant TU/C/000023. VK is supported in part by the Alan Turing Institute under the EPSRC grant EP/N510129/1. PHS and PD acknowledges support from the Royal Academy of Engineering under the Research Chair and Senior Research Fellowships scheme, EPSRC/MURI grant EP/N019474/1 and FiveAI.

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

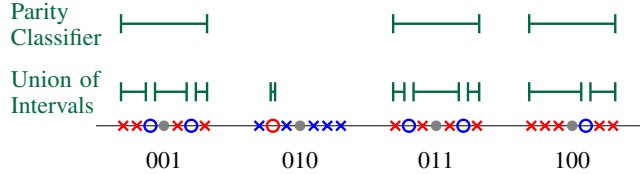 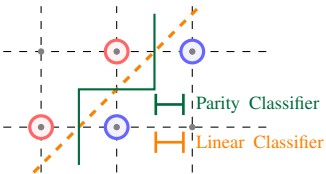

(a) Both Parity and Union of Interval classifier predicts red if inside any green interval and blue if outside all intervals. The ×-es are correctly labelled and the ○-es are mis-labelled points. Reference integer points on the line labelled in *binary*.

(b) Robust generalization needs more complex boundaries

Figure 6: Visualization of the distribution and classifiers used in the Proof of Theorem 2 and 3. The Red and Blue indicate the two classes.

## A    REPRESENTATION LEARNING AND ADVERSARIAL ROBUSTNESS

In this section we present another result to show that there exists a data distribution where proper representation is necessary even in the absence of label noise for small adversarial error as well as small test error whereas another representation can provide low test error but necessarily have large adversarial error. Interestingly, the representation that can achieve small adversarial error can look visually more complex due to larger number of distinct linear regions in its decision boundary. However, statistically it will have a smaller VC dimension than its counterpart.

We first present the theorems with a proof sketch for ease of understanding and the more detailed proofs in Appendix B.

**Theorem 3.** *For some universal constant c, and any $0 < \gamma_0 < 1/\sqrt{2}$, there exists a family of distributions $\mathcal{D}$ defined on $\mathcal{X} \times \{0,1\}$ where $\mathcal{X} \subseteq \mathbb{R}^2$ such that for all distributions $\mathcal{P} \in \mathcal{D}$, and denoting by $\mathcal{S}_m = \{(\mathbf{x}_1, y_1), \cdots, (\mathbf{x}_m, y_m)\}$ a sample of size $m$ drawn i.i.d. from $\mathcal{P}$,*

*(i) For any $m \geq 0$, $\mathcal{S}_m$ is linearly separable i.e., $\forall(\mathbf{x}_i, y_i) \in \mathcal{S}_m$, there exist $\mathbf{w} \in \mathbb{R}^2, w_0 \in \mathbb{R}$ s.t. $y_i(\mathbf{w}^\top \mathbf{x}_i + w_0) \geq 0$. Furthermore, for every $\gamma > \gamma_0$, any linear separator $f$ that perfectly fits the training data $\mathcal{S}_m$ has $\mathcal{R}_{\mathrm{Adv},\gamma}(f; \mathcal{P}) \geq 0.0005$, even though $\mathcal{R}(f; \mathcal{P}) \to 0$ as $m \to \infty$.*

*(ii) There exists a function class $\mathcal{H}$ such that for some $m \in O(\log(\delta^{-1}))$, any $h \in \mathcal{H}$ that perfectly fits the $\mathcal{S}_m$, satisfies with probability at least $1 - \delta$, $\mathcal{R}(h; \mathcal{P}) = 0$ and $\mathcal{R}_{\mathrm{Adv},\gamma}(h; \mathcal{P}) = 0$, for any $\gamma \in [0, \gamma_0 + 1/8]$.*

A complete proof of this result appears in Appendix B.2, but first, we provide a sketch of the key idea here.The distributions in family $\mathcal{D}$ will be supported on balls of radius at most $1/\sqrt{2}$ on the integer lattice in $\mathbb{R}^2$. The *true* class label for any point $\mathbf{x}$ is provided by the parity of $a + b$, where $(a, b)$ is the lattice point closest to $\mathbf{x}$. However, the distributions in $\mathcal{D}$ are chosen to be such that there is also a linear classifier that can separate these classes, e.g. a distribution only supported on balls centered at the points $(a, a)$ and $(a, a + 1)$ for some integer $a$ (See Figure 6b). *Visually* learning the classification problem using the parity of $a + b$ results in a seemingly more complex decision boundary, a point that has been made earlier regarding the need for more complex boundaries to achieve adversarial robustness (Nakkiran, 2019; Degwekar et al., 2019). However, it is worth noting that this complexity is not rooted in any *statistical theory*, e.g. the VC dimension of the classes considered in Theorem 3 is essentially the same (even lower for $\mathcal{H}$ by 1). This *visual* complexity arises purely due to the fact that the linear classifier looks at a geometric representation of the data whereas the parity classifier looks at the binary representation of the sum of the nearest integer of the coordinates. In the case of neural networks, recent works (Kamath et al., 2020) have indeed provided empirical results to support that excessive invariance (eg. rotation invariance) increases adversarial error.

### REPRESENTATION LEARNING IN THE PRESENCE OF LABEL NOISE

Here we restate Theorem 2 and present a proof sketch for the result.

**Theorem 2.**    *For any $n \in \mathbb{Z}_+$, there exists a family of distributions $\mathcal{D}^n$ over $\mathbb{R} \times \{0,1\}$ and function classes $\mathcal{C}, \mathcal{H}$, such that for any $\mathcal{P}$ from $\mathcal{D}^n$, and for any $0 < \gamma < 1/4$, and $\eta \in (0, 1/2)$ if*

$\mathcal{S}_m = \{(\mathbf{x}_i, y_i)\}_{i=1}^m$ *denotes a sample of size* $m$ *drawn from* $\mathcal{P}$ *where*

$$m = O\left(\max\left\{n\log\frac{n}{\delta}\left(\frac{(1-\eta)}{(1-2\eta)^2}+1\right), \frac{n}{\eta\gamma^2}\log\left(\frac{n}{\gamma\delta}\right)\right\}\right)$$

*and if* $\mathcal{S}_{m,\eta}$ *denotes the sample where each label is flipped independently with probability* $\eta$.

*(i) the classifier* $c \in \mathcal{C}$ *that minimizes the training error on* $\mathcal{S}_{m,\eta}$, *has* $\mathcal{R}(c;\mathcal{P}) = 0$ *and* $\mathcal{R}_{\mathrm{Adv},\gamma}(c;\mathcal{P}) = 0$ *for* $0 \le \gamma < 1/4$.

*(ii) there exist* $h \in \mathcal{H}$, $h$ *has zero training error on* $\mathcal{S}_{m,\eta}$, *and* $\mathcal{R}(h;\mathcal{P}) = 0$. *However, for any* $\gamma > 0$, *and for any* $h \in \mathcal{H}$ *with zero training error on* $\mathcal{S}_{m,\eta}$, $\mathcal{R}_{\mathrm{Adv},\gamma}(h;\mathcal{P}) \ge 0.1$.

*Furthermore, the required* $c \in \mathcal{C}$ *and* $h \in \mathcal{H}$ *above can be computed in* $O\left(\mathrm{poly}(n), \mathrm{poly}\left(\frac{1}{\frac{1}{2}-\eta}\right), \mathrm{poly}\left(\frac{1}{\delta}\right)\right)$ *time.*

We sketch the proof here and present the complete the proof in Appendix B.3; as in Appendix A we will make use of parity functions, though the key point is the representations used. Let $\mathcal{X} = [0, N]$, where $N = 2^n$, we consider distributions that are supported on intervals $(i - 1/4, i + 1/4)$ for $i \in \{1, \ldots, N-1\}$ (See Figure 6a), but any such distribution will only have a small number, $O(n)$, intervals on which it is supported. The *true* class label is given by a function that depends on the parity of some hidden subsets $S$ of bits in the bit-representation of the closest integer $i$, e.g. as in Figure 6a if $S = \{0, 2\}$, then only the least significant and the third least significant bit of $i$ are examined and the class label is 1 if an odd number of them are 1 and 0 otherwise. Despite the noise, the *correct* label on any interval can be guessed by using the majority vote and as a result, the correct parity learnt using Gaussian elimination. (This corresponds to the class $\mathcal{C}$ in Theorem 2.) On the other hand it is also possible to learn the function as a union of intervals, i.e. find intervals, $I_1, I_2, \ldots, I_k$ such that any point that lies in one of these intervals is given the label 1 and any other point is given the label 0. By choosing intervals carefully, it is possible to fit *all the training data*, including noisy examples, but yet not compromise on *test accuracy* (Fig. 6a). Such a classifier, however, will be vulnerable to adversarial examples by applying Theorem 1. A classifier such as union of intervals ($\mathcal{H}$ in Theorem 2) is translation-invariant, whereas the parity classifier is not. This suggests that using classifiers, such as neural networks, that are designed to have too many built-in invariances might hurt its robustness accuracy.

# B PROOFS FOR SECTION 2

In this section, we present the formal proofs to the theorems stated in Section 2.

## B.1 PROOF OF THEOREM 1

**Theorem 1.** *Let* $c$ *be the target classifier, and let* $\mathcal{D}$ *be a distribution over* $(\mathbf{x}, y)$, *such that* $y = c(\mathbf{x})$ *in its support. Using the notation* $\mathbb{P}_{\mathcal{D}}[A]$ *to denote* $\mathbb{P}_{(\mathbf{x},y)\sim\mathcal{D}}[\mathbf{x} \in A]$ *for any measurable subset* $A \subseteq \mathbb{R}^d$, *suppose that there exist* $c_1 \ge c_2 > 0$, $\rho > 0$, *and a finite set* $\zeta \subset \mathbb{R}^d$ *satisfying*

$$\mathbb{P}_{\mathcal{D}}\left[\bigcup_{\mathbf{s}\in\zeta}\mathcal{B}_\rho^p(\mathbf{s})\right] \ge c_1 \quad and \quad \forall \mathbf{s} \in \zeta,\ \mathbb{P}_{\mathcal{D}}\left[\mathcal{B}_\rho^p(\mathbf{s})\right] \ge \frac{c_2}{|\zeta|} \tag{3}$$

*where* $\mathcal{B}_\rho^p(\mathbf{s})$ *represents a* $\ell_p$-*ball of radius* $\rho$ *around* $\mathbf{s}$. *Further, suppose that each of these balls contain points from a single class i.e. for all* $\mathbf{s} \in \zeta$, *for all* $\mathbf{x}, \mathbf{z} \in \mathcal{B}_\rho^p(\mathbf{s})$: $c(\mathbf{x}) = c(\mathbf{z})$.

*Let* $\mathcal{S}_m$ *be a dataset of* $m$ *i.i.d. samples drawn from* $\mathcal{D}$, *which subsequently has each label flipped independently with probability* $\eta$. *For any classifier* $f$ *that* perfectly *fits the training data* $\mathcal{S}_m$ *i.e.* $\forall\ \mathbf{x}, y \in \mathcal{S}_m$, $f(\mathbf{x}) = y$, $\forall \delta > 0$ *and* $m \ge \frac{|\zeta|}{\eta c_2}\log\left(\frac{|\zeta|}{\delta}\right)$, *with probability at least* $1 - \delta$, $\mathcal{R}_{\mathrm{Adv},2\rho}(f;\mathcal{D}) \ge c_1$.

*Proof of Theorem 1.* From (3), for any $\zeta$ and $s \in \zeta$,

$$\mathbb{P}_{(\mathbf{x},y)\sim\mathcal{D}}\left[\mathbf{x} \in \mathcal{B}_\rho(s)\right] \ge \frac{c_2}{|\zeta|}$$

As the sampling of the point and the injection of label noise are independent events,

$$\mathbb{P}_{(\mathbf{x},y)\sim\mathcal{D}}\left[\mathbf{x}\in\mathcal{B}_\rho\left(s\right)\wedge\mathbf{x}\text{ gets mislabelled}\right]\geq\frac{c_2\eta}{|\zeta|}$$

Thus,

$$\mathbb{P}_{\mathcal{S}_m\sim\mathcal{D}^m}\left[\exists\left(\mathbf{x},y\right)\in\mathcal{S}_m:\mathbf{x}\in\mathcal{B}_\rho\left(s\right)\wedge\mathbf{x}\text{ is mislabelled}\right]\geq 1-\left(1-\frac{c_2\eta}{|\zeta|}\right)^m$$

$$\geq 1-\exp\left(\frac{-c_2\eta m}{|\zeta|}\right)$$

Substituting $m\geq\frac{|\zeta|}{\eta c_2}\log\left(\frac{|\zeta|}{\delta}\right)$ and applying the union bound over all $s\in\zeta$, we get

$$\mathbb{P}_{\mathcal{S}_m\sim\mathcal{D}^m}\left[\forall s\in\zeta,\ \exists\left(\mathbf{x},y\right)\in\mathcal{S}_m:\mathbf{x}\in\mathcal{B}_\rho\left(s\right)\wedge\mathbf{x}\text{ is mislabelled}\right]\geq 1-\delta \tag{4}$$

As for all $\mathbf{s}\in\mathbb{R}^d$ and $\forall\mathbf{x},\mathbf{z},\in\mathcal{B}_\rho^p\left(\mathbf{s}\right),\ \|\mathbf{x}-\mathbf{z}\|_p\leq 2\rho$, we have that

$$\mathcal{R}_{\text{Adv},2\rho}(f;\mathcal{D})=\mathbb{P}_{\mathcal{S}_m\sim\mathcal{D}^m}\left[\mathbb{P}_{(\mathbf{x},y)\sim\mathcal{D}}\left[\exists\mathbf{z}\in\mathcal{B}_{2\rho}\left(\mathbf{x}\right)\ \wedge y\neq f\left(\mathbf{z}\right)\right]\right]$$

$$\geq\mathbb{P}_{\mathcal{S}_m\sim\mathcal{D}^n}\left[\mathbb{P}_{(\mathbf{x},y)\sim\mathcal{D}}\left[\mathbf{x}\in\bigcup_{s\in\zeta}\mathcal{B}_\rho^p\left(s\right)\wedge\{\exists\mathbf{z}\in\mathcal{B}_{2\rho}\left(\mathbf{x}\right):y\neq f\left(\mathbf{z}\right)\}\right]\right]$$

$$\overset{(1)}{=}\mathbb{P}_{\mathcal{S}_m\sim\mathcal{D}^n}\left[\mathbb{P}_{(\mathbf{x},y)\sim\mathcal{D}}\left[\exists\mathbf{s}\in\zeta:\mathbf{x}\in\mathcal{B}_\rho^p\left(s\right)\wedge\{\exists\mathbf{z}\in\mathcal{B}_{2\rho}\left(\mathbf{x}\right):y\neq f\left(\mathbf{z}\right)\}\right]\right]$$

$$\geq\mathbb{P}_{\mathcal{S}_m\sim\mathcal{D}^m}\left[\mathbb{P}_{(\mathbf{x},y)\sim\mathcal{D}}\left[\exists\mathbf{s}\in\zeta:\mathbf{x}\in\mathcal{B}_\rho^p\left(s\right)\wedge\{\exists\mathbf{z}\in\mathcal{B}_\rho\left(\mathbf{s}\right):y\neq f\left(\mathbf{z}\right)\}\right]\right]$$

$$\overset{(2)}{=}\mathbb{P}_{\mathcal{S}_m\sim\mathcal{D}^m}\left[\mathbb{P}_{(\mathbf{x},y)\sim\mathcal{D}}\left[\exists\mathbf{s}\in\zeta:\mathbf{x}\in\mathcal{B}_\rho^p\left(s\right)\wedge\{\exists\mathbf{z}\in\mathcal{B}_\rho\left(\mathbf{s}\right):c\left(\mathbf{z}\right)\neq f\left(\mathbf{z}\right)\}\right]\right]$$

$$\overset{(3)}{=}\mathbb{P}_{(\mathbf{x},y)\sim\mathcal{D}}\left[\mathbf{x}\in\bigcup_{s\in\zeta}\mathcal{B}_\rho^p\left(s\right)\right]\quad\text{w.p. atleast }1-\delta$$

$$\geq c_1\quad\text{w.p. }1-\delta$$

where $c$ is the true concept for the distribution $\mathcal{D}$. Equality (1) follows from the fact that the event $\mathbf{x}\in\bigcup_{s\in\zeta}\mathcal{B}_\rho^p\left(s\right)$ is equivalent to the event that there exists an $i\in\{1,..,|\zeta|\}$ such that $\mathbf{x}\in\mathcal{B}_\rho^p\left(s_i\right)$. Equality (2) follows from the assumptions that each of the balls around $\mathbf{s}\in\zeta$ are pure in their labels. Equality (3) follows from 4 by using the $\mathbf{x}$ that is guaranteed to exist in the ball around $\mathbf{s}$ and be mis-labelled with probability atleast $1-\delta$. To see how, replace the $\mathbf{x}$ in Assumption 4 with $\mathbf{z}$. The last equality follows from the first Assumption in 3. $\qquad\square$

## B.2  PROOFS OF APPENDIX A

**Theorem 3.** *For some universal constant $c$, and any $0<\gamma_0<1/\sqrt{2}$, there exists a family of distributions $\mathcal{D}$ defined on $\mathcal{X}\times\{0,1\}$ where $\mathcal{X}\subseteq\mathbb{R}^2$ such that for all distributions $\mathcal{P}\in\mathcal{D}$, and denoting by $\mathcal{S}_m=\{(\mathbf{x}_1,y_1),\cdots,(\mathbf{x}_m,y_m)\}$ a sample of size $m$ drawn i.i.d. from $\mathcal{P}$,*

*(i) For any $m\geq 0$, $\mathcal{S}_m$ is linearly separable i.e., $\forall(\mathbf{x}_i,y_i)\in\mathcal{S}_m$, there exist $\mathbf{w}\in\mathbb{R}^2,w_0\in\mathbb{R}$ s.t. $y_i\left(\mathbf{w}^\top\mathbf{x}_i+w_0\right)\geq 0$. Furthermore, for every $\gamma>\gamma_0$, any linear separator $f$ that perfectly fits the training data $\mathcal{S}_m$ has $\mathcal{R}_{\text{Adv},\gamma}(f;\mathcal{P})\geq 0.0005$, even though $\mathcal{R}(f;\mathcal{P})\to 0$ as $m\to\infty$.*

*(ii) There exists a function class $\mathcal{H}$ such that for some $m\in O(\log(\delta^{-1}))$, any $h\in\mathcal{H}$ that perfectly fits the $\mathcal{S}_m$, satisfies with probability at least $1-\delta$, $\mathcal{R}(h;\mathcal{P})=0$ and $\mathcal{R}_{\text{Adv},\gamma}(h;\mathcal{P})=0$, for any $\gamma\in[0,\gamma_0+1/8]$.*

*Proof of Theorem 3.* We define a family of distribution $\mathcal{D}$, such that each distribution in $\mathcal{D}$ is supported on balls of radius $r$ around $(i,i)$ and $(i+1,i)$ for positive integers $i$. Either all the balls around $(i,i)$ have the labels 1 and the balls around $(i+1,i)$ have the label 0 or vice versa. Figure 6b shows an example where the colors indicate the label.

Formally, for $r > 0$, $k \in \mathbb{Z}_+$, the $(r, k)$-1 bit parity class conditional model is defined over $(x, y) \in \mathbb{R}^2 \times \{0, 1\}$ as follows. First, a label $y$ is sampled uniformly from $\{0, 1\}$, then and integer $i$ is sampled uniformly from the set $\{1, \cdots, k\}$ and finally $\mathbf{x}$ is generated by sampling uniformly from the $\ell_2$ ball of radius $r$ around $(i + y, i)$.

In Lemma 1 we first show that a set of $m$ points sampled iid from any distribution as defined above for $r < \frac{1}{2\sqrt{2}}$ is with probability 1 linear separable for any $m$. In addition, standard VC bounds show that any linear classifier that separates $S_m$ for large enough $m$ will have small test error. Lemma 1 also proves that there exists a range of $\gamma, r$ such that for any distribution defined with $r$ in that range, though it is possible to obtain a linear classifier with 0 training and test error, the minimum adversarial risk will be bounded from 0.

However while it is possible to obtain a linear classifier with 0 test error, all such linear classifiers has a large adversarial vulnerability. In Lemma 2, we show that there exists a different representation for this problem, which also achieves zero training and test error and in addition has zero adversarial risk for a range of $r, \gamma$ where the linear classifier's adversarial error was atleast a constant.

$\square$

**Lemma 1** (Linear Classifier). *There exists universal constants $\gamma_0, \rho$, such that for any perturbation $\gamma > \gamma_0$, radius $r \geq \rho$, and $k \in \mathbb{Z}_+$, the following holds. Let $\mathcal{D}$ be the family of $(r, k)$- 1-bit parity class conditional model, $\mathcal{P} \in \mathcal{D}$ and $S_n = \{(\mathbf{x}_1, y_1), \cdots, (\mathbf{x}_n, y_1)\}$ be a set of $n$ points sampled i.i.d. from $\mathcal{P}$.*

1) *For any $n > 0$, $S_n$ is linearly separable with probability 1 i.e. there exists a $h : (\mathbf{w}, w_0)$, $\mathbf{w} \in \mathbb{R}^2, w_0 \in \mathbb{R}$ such that the linear hyperplane $\mathbf{x} \to \mathbf{w}^\top \mathbf{x} + w_0$ separates $S_n$ with probability 1:*

$$\forall (\mathbf{x}, y) \in S_n \quad z(\mathbf{w}^\top \mathbf{x} + w_0) > 0 \quad \text{where } z = 2y - 1$$

2) *Further there exists an universal constant $c$ such that for any $\epsilon, \delta > 0$ with probability $1 - \delta$ for any $S_n$ with $n = c\frac{1}{\epsilon^2} \log \frac{1}{\delta}$, any linear classifier $\tilde{h}$ that separates $S_n$ has $\mathcal{R}(\tilde{h}; \mathcal{P}) \leq \epsilon$.*
3) *Let $h : (\mathbf{w}, w_0)$ be any linear classifier that has $\mathcal{R}(h; \mathcal{P}_P) = 0$. Then, $\mathcal{R}_{\text{Adv},\gamma}(h; \mathcal{P}) > 0.0005$.*

We will prove the first part for any $r < \frac{1}{2\sqrt{2}}$ by constructing a $\mathbf{w}, w_0$ such that it satisfies the constraints of linear separability. Let $\mathbf{w} = (1, -1)$, $w_0 = -0.5$. Consider any point $(\mathbf{x}, y) \in S_n$ and $z = 2y - 1$. Converting to the polar coordinate system there exists a $\theta \in [0, 2\pi]$, $j \in [0, \cdots, k]$ such that $\mathbf{x} = \left(j + \frac{z+1}{2} + r\cos(\theta), j + r\sin(\theta)\right)$

$$
\begin{aligned}
z(\mathbf{w}^\top \mathbf{x} + w_0) &= z\left(j + \frac{z+1}{2} + r\cos(\theta) - j - r\sin(\theta) - 0.5\right) \quad \mathbf{w} = (1, -1)^\top \\
&= z\left(\frac{z}{2} + 0.5 + r\cos(\theta) - r\sin(\theta) - 0.5\right) \\
&= \frac{1}{2} + zr(\cos(\theta) - \sin(\theta)) \quad |\cos(\theta) - \sin(\theta)| < \sqrt{2}, \ z \in \{-1, 1\} \\
&> \frac{1}{2} - r\sqrt{2} \\
&> 0 \quad r < \frac{1}{2\sqrt{2}}
\end{aligned}
$$

Part 2 follows with simple VC bounds of linear classifiers.

Let the universal constants $\gamma_0, \rho$ be 0.02 and $\frac{1}{2\sqrt{2}} - 0.008$ respectively. Note that there is nothing special about this constants except that *some* constant is required to bound the adversarial risk away from 0. Now, consider a distribution $\mathcal{P}$ 1-bit parity model such that the radius of each ball is atleast $\rho$. This is smaller than $\frac{1}{2\sqrt{2}}$ and thus satisfies the linear separability criterion.

Consider $h$ to be a hyper-plane that has 0 test error. Let the $\ell_2$ radius of adversarial perturbation be $\gamma > \gamma_0$. The region of each circle that will be vulnerable to the attack will be a circular segment with the chord of the segment parallel to the hyper-plane. Let the minimum height of all such circular

segments be $r_0$. Thus, $\mathcal{R}_{\mathrm{Adv},\gamma}(h;\mathcal{P})$ is greater than the mass of the circular segment of radius $r_0$. Let the radius of each ball in the support of $\mathcal{P}$ be $r$.

Using the fact that $h$ has zero test error; and thus classifies the balls in the support of $\mathcal{P}$ correctly and simple geometry

$$\frac{1}{\sqrt{2}} \geq r + (\gamma - r_0) + r$$

$$r_0 \geq 2r + \gamma - \frac{1}{\sqrt{2}} \tag{5}$$

To compute $\mathcal{R}_{\mathrm{Adv},\gamma}(h;\mathcal{P})$ we need to compute the ratio of the area of a circular segment of height $r_0$ of a circle of radius $r$ to the area of the circle. The ratio can be written

$$A\left(\frac{r_0}{r}\right) = \frac{cos^{-1}\left(1 - \frac{r_0}{r}\right) - \left(1 - \frac{r_0}{r}\right)\sqrt{2\frac{r_0}{r} - \frac{r_0^2}{r^2}}}{\pi} \tag{6}$$

As (6) is increasing with $\frac{r_0}{r}$, we can evaluate

$$\frac{r_0}{r} \geq \frac{2r - \frac{1}{\sqrt{2}} + \gamma}{r} \qquad \text{Using (5)}$$

$$\geq 2 - \frac{\frac{1}{\sqrt{2}} - 0.02}{r} \qquad \gamma > \gamma_0 = 0.02$$

$$\geq 2 - \frac{\frac{1}{\sqrt{2}} - 0.02}{\frac{1}{\sqrt{2}} - 0.008} > 0.01 \qquad r > \rho = \frac{1}{2\sqrt{2}} - 0.008$$

Substituting $\frac{r_0}{r} > 0.01$ into Eq. (6), we get that $A\left(\frac{r_0}{r}\right) > 0.0005$. Thus, for all $\gamma > 0.02$, we have $\mathcal{R}_{\mathrm{Adv},\gamma}(h;\mathcal{P}) > 0.0005$.

**Lemma 2** (Robustness of parity classifier). *There exists a concept class $\mathcal{H}$ such that for any $\gamma \in \left[\gamma_0, \gamma_0 + \frac{1}{8}\right]$, $k \in \mathbb{Z}_+$, $\mathcal{P}$ being the corresponding $(\rho, k)$ 1-bit parity class distribution where $\rho, \gamma_0$ are the same as in Lemma 1 there exists $g \in \mathcal{H}$ such that*

$$\mathcal{R}(g;\mathcal{P}) = 0 \qquad \mathcal{R}_{\mathrm{Adv},\gamma}(g;\mathcal{P}) = 0$$

*Proof of Lemma 2.* We will again provide a proof by construction. Consider the following class of concepts $\mathcal{H}$ such that $g_b \in \mathcal{H}$ is defined as

$$g\left((x_1, x_2)^\top\right) = \begin{cases} 1 & \text{if } [x_1] + [x_2] = b \,(\mathrm{mod}\,2) \\ 1 - b & \text{o.w.} \end{cases} \tag{7}$$

where $[x]$ rounds $x$ to the nearest integer and $b \in \{0, 1\}$. In Figure 6b, the green staircase-like classifier belongs to this class. Consider the classifier $g_1$. Note that by construction $\mathcal{R}(g_1;\mathcal{P}) = 0$. The decision boundary of $g_1$ that are closest to a ball in the support of $\mathcal{P}$ centered at $(a, b)$ are the lines $x = a \pm 0.5$ and $y = b \pm 0.5$.

As $\gamma < \gamma_0 + \frac{1}{8}$, the adversarial perturbation is upper bounded by $\frac{1}{50} + \frac{1}{8}$. The radius of the ball is upper bounded by $\frac{1}{2\sqrt{2}}$, and as we noted the center of the ball is at a distance of $0.5$ from the decision boundary. If the sum of the maximum adversarial perturbation and the maximum radius of the ball is less than the minimum distance of the center of the ball from the decision boundary, then the adversarial error is $0$. Substituting the values,

$$\frac{1}{50} + \frac{1}{8} + \frac{1}{2\sqrt{2}} < 0.499 < \frac{1}{2}$$

This completes the proof. $\qquad\qquad\square$

### B.3 Proof of Appendix A

**Theorem 2.**    *For any $n \in \mathbb{Z}_+$, there exists a family of distributions $\mathcal{D}^n$ over $\mathbb{R} \times \{0,1\}$ and function classes $\mathcal{C}, \mathcal{H}$, such that for any $\mathcal{P}$ from $\mathcal{D}^n$, and for any $0 < \gamma < 1/4$, and $\eta \in (0, 1/2)$ if $\mathcal{S}_m = \{(\mathbf{x}_i, y_i)\}_{i=1}^m$ denotes a sample of size $m$ drawn from $\mathcal{P}$ where*

$$m = O\left(\max\left\{ n \log \frac{n}{\delta} \left( \frac{(1-\eta)}{(1-2\eta)^2} + 1 \right), \frac{n}{\eta\gamma^2} \log\left(\frac{n}{\gamma\delta}\right) \right\}\right)$$

*and if $\mathcal{S}_{m,\eta}$ denotes the sample where each label is flipped independently with probability $\eta$.*

*(i)  the classifier $c \in \mathcal{C}$ that minimizes the training error on $\mathcal{S}_{m,\eta}$, has $\mathcal{R}(c; \mathcal{P}) = 0$ and $\mathcal{R}_{\mathrm{Adv},\gamma}(c; \mathcal{P}) = 0$ for $0 \leq \gamma < 1/4$.*

*(ii) there exist $h \in \mathcal{H}$, $h$ has zero training error on $\mathcal{S}_{m,\eta}$, and $\mathcal{R}(h; \mathcal{P}) = 0$. However, for any $\gamma > 0$, and for any $h \in \mathcal{H}$ with zero training error on $\mathcal{S}_{m,\eta}$, $\mathcal{R}_{\mathrm{Adv},\gamma}(h; \mathcal{P}) \geq 0.1$.*

*Furthermore, the required $c \in \mathcal{C}$ and $h \in \mathcal{H}$ above can be computed in $O\left(\mathrm{poly}\,(n), \mathrm{poly}\left(\frac{1}{\frac{1}{2}-\eta}\right), \mathrm{poly}\left(\frac{1}{\delta}\right)\right)$ time.*

*Proof of Theorem 2.*  We will provide a constructive proof to this theorem by constructing a distribution $\mathcal{D}$, two concept classes $\mathcal{C}$ and $\mathcal{H}$ and provide the ERM algorithms to learn the concepts and then use Lemma 3 and 4 to complete the proof.

**Distribution:** Consider the family of distribution $\mathcal{D}^n$ such that $\mathcal{D}_{S,\zeta} \in \mathcal{D}^n$ is defined on $\mathcal{X}_\zeta \times \{0,1\}$ for $S \subseteq \{1, \cdots, n\}, \zeta \subseteq \{1, \cdots, 2^n - 1\}$ such that the support of $\mathcal{X}_\zeta$ is a union of intervals.

$$\mathrm{supp}\,(\mathcal{X})_\zeta = \bigcup_{j \in \zeta} I_j \text{ where } I_j := \left(j - \frac{1}{4}, j + \frac{1}{4}\right) \tag{8}$$

We consider distributions with a relatively small support i.e. where $|\zeta| = O(n)$. Each sample $(\mathbf{x}, y) \sim \mathcal{D}_{S,\zeta}$ is created by sampling $\mathbf{x}$ uniformly from $\mathcal{X}_\zeta$ and assigning $y = c_S(\mathbf{x})$ where $c_S \in \mathcal{C}$ is defined below (9). We define the family of distributions $\mathcal{D} = \bigcup_{n \in \mathbb{Z}_+} \mathcal{D}^n$. Finally, we create $\mathcal{D}_{S,\zeta}^\eta$ -a noisy version of $\mathcal{D}_{S,\zeta}$, by flipping $y$ in each sample $(x, y)$ with probability $\eta < \frac{1}{2}$. Samples from $\mathcal{D}_{S,\zeta}$ can be obtained using the example oracle $\mathrm{EX}\,(\mathcal{D}_{S,\zeta})$ and samples from the noisy distribution can be obtained through the noisy oracle $\mathrm{EX}^\eta\,(\mathcal{D}_{S,\zeta})$

**Concept Class $\mathcal{C}$:** We define the concept class $\mathcal{C}^n$ of concepts $c_S : [0, 2^n] \to \{0,1\}$ such that

$$c_S(\mathbf{x}) = \begin{cases} 1, & \text{if } (\langle [\mathbf{x}] \rangle_b \text{ XOR } S) \text{ is odd.} \\ 0 & \text{o.w.} \end{cases} \tag{9}$$

where $[\cdot] : \mathbb{R} \to \mathbb{Z}$ rounds a decimal to its nearest integer, $\langle \cdot \rangle_b : \{0, \cdots, 2^n\} \to \{0,1\}^n$ returns the binary encoding of the integer, and $(\langle [\mathbf{x}] \rangle_b \text{ XOR } S) = \sum_{j \in S} \langle [x] \rangle_b [j] \mod 2$. $\langle [x] \rangle_b [j]$ is the $j^{th}$ least significant bit in the binary encoding of the nearest integer to $\mathbf{x}$. It is essentially the class of parity functions defined on the bits corresponding to the indices in $S$ for the binary encoding of the nearest integer to $\mathbf{x}$. For example, as in Figure 6a if $S = \{0, 2\}$, then only the least significant and the third least significant bit of $i$ are examined and the class label is 1 if an odd number of them are 1 and 0 otherwise.

**Concept Class $\mathcal{H}$:** Finally, we define the concept class $\mathcal{H} = \bigcup_{k=1}^\infty \mathcal{H}_k$ where $\mathcal{H}_k$ is the class of union of $k$ intervals on the real line $\mathcal{H}^k$. Each concept $h_I \in \mathcal{H}^k$ can be written as a set of $k$ disjoint intervals $I = \{I_1, \cdots, I_k\}$ on the real line i.e. for $1 \leq j \leq k$, $I_j = [a, b]$ where $0 \leq a \leq b$ and

$$h_I(\mathbf{x}) = \begin{cases} 1 & \text{if } \mathbf{x} \in \bigcup_j I_j \\ 0 & \text{o.w.} \end{cases} \tag{10}$$

Now, we look at the algorithms to learn the concepts from $\mathcal{C}$ and $\mathcal{H}$ that minimize the train error. Both of the algorithms will use a majority vote to determine the correct (de-noised) label for each interval, which will be necessary to minimize the test error. The intuition is that if we draw a sufficiently large

number of samples, then the majority of samples on each interval will have the correct label with a high probability.

Lemma 3 proves that there exists an algorithm $\mathcal{A}$ such that $\mathcal{A}$ draws $m = O\left(|\zeta|^2 \frac{(1-\eta)}{(1-2\eta)^2} \log \frac{|\zeta|}{\delta}\right)$ samples from the noisy oracle $\mathrm{EX}^\eta(\mathcal{D}_{s,\zeta})$ and with probability $1 - \delta$ where the probability is over the randomization in the oracle, returns $f \in \mathcal{C}$ such that $\mathcal{R}(f; \mathcal{D}_{S,\zeta}) = 0$ and $\mathcal{R}_{\mathrm{Adv},\gamma}(f; \mathcal{D}_{S,\zeta}) = 0$ for all $\gamma < \frac{1}{4}$. As Lemma 3 states, the algorithm involves gaussian elimination over $|\zeta|$ variables and $|\zeta|$ majority votes (one in each interval) involving a total of $m$ samples. Thus the algorithm runs in $O\left(\mathrm{poly}\,(m) + \mathrm{poly}\,(|\zeta|)\right)$ time. Replacing the complexity of $m$ and the fact that $|\zeta| = O(n)$, the complexity of the algorithm is $O\left(\mathrm{poly}\left(n, \frac{1}{1-2\eta}, \frac{1}{\delta}\right)\right)$.

Lemma 4 proves that there exists an algorithm $\widetilde{A}$ such that $\widetilde{A}$ draws

$$m > \max\left\{2\,|\zeta|^2 \log \frac{2\,|\zeta|}{\delta}\left(8\frac{(1-\eta)}{(1-2\eta)^2} + 1\right), \frac{0.1\,|\zeta|}{\eta\gamma^2} \log\left(\frac{0.1\,|\zeta|}{\gamma\delta}\right)\right\}$$

samples and returns $h \in \mathcal{H}$ such that $h$ has 0 training error, 0 test error and an adversarial test error of atleast 0.1. We can replace $|\zeta| = O(n)$ to get the required bound on $m$ in the theorem. The algorithm to construct $h$ visits every point atmost twice - once during the construction of the intervals using majority voting, and once while accommodating for the mislabelled points. Replacing the complexity of $m$, the complexity of the algorithm is $O\left(\mathrm{poly}\left(n, \frac{1}{1-2\eta}, \frac{1}{\gamma}, \frac{1}{\delta}\right)\right)$. This completes the proof. $\qquad\square$

**Lemma 3** (Parity Concept Class). *There exists a learning algorithm $\mathcal{A}$ such that given access to the noisy example oracle $\mathrm{EX}^\eta(\mathcal{D}_{S,\zeta})$, $\mathcal{A}$ makes $m = O\left(|\zeta|^2 \frac{(1-\eta)}{(1-2\eta)^2} \log \frac{|\zeta|}{\delta}\right)$ calls to the oracle and returns a hypothesis $f \in \mathcal{C}$ such that with probability $1 - \delta$, we have that $\mathcal{R}(f; \mathcal{D}_{S,\zeta}) = 0$ and $\mathcal{R}_{\mathrm{Adv},\gamma}(f; \mathcal{D}_{S,\zeta}) = 0$ for all $\gamma < \frac{1}{4}$.*

*Proof.* The algorithm $\mathcal{A}$ works as follows. It makes $m$ calls to the oracle $\mathrm{EX}(\mathcal{D}_s^m)$ to obtain a set of points $\{(x_1, y_1), \cdots, (x_m, y_m)\}$ where $m \geq 2\,|\zeta|^2 \log \frac{2|\zeta|}{\delta}\left(8\frac{(1-\eta)}{(1-2\eta)^2} + 1\right)$. Then, it replaces each $x_i$ with $[x_i]$ ($[\cdot]$ rounds a decimal to the nearest integer) and then removes duplicate $x_i$s by preserving the most frequent label $y_i$ associated with each $x_i$. For example, if $\mathcal{S}_5 = \{(2.8, 1), (2.9, 0), (3.1, 1), (3.2, 1), (3.9, 0)\}$ then after this operation, we will have $\{(3, 1), (4, 0)\}$.

As $m \geq 2\,|\zeta|^2 \log \frac{2|\zeta|}{\delta}\left(8\frac{(1-\eta)}{(1-2\eta)^2} + 1\right)$, using $\delta_2 = \frac{\delta}{2}$ and $k = \frac{8(1-\eta)}{(1-2\eta)^2} \log \frac{2|\zeta|}{\delta}$ in Lemma 5 guarantees that with probability $1 - \frac{\delta}{2}$, each interval will have atleast $\frac{8(1-\eta)}{(1-2\eta)^2} \log \frac{2|\zeta|}{\delta}$ samples.

Then for any specific interval, using $\delta_1 = \frac{2|\zeta|}{\delta}$ in Lemma 6 guarantees that with probability atleast $1 - \frac{2|\zeta|}{\delta}$, the majority vote for the label in that interval will succeed in returning the de-noised label. Applying a union bound over all $|\zeta|$ intervals, will guarantee that with probability atleast $1 - \delta$, the majority label of every interval will be the denoised label.

Now, the problem reduces to solving a parity problem on this reduced dataset of $|\zeta|$ points (after denoising, all points in that interval can be reduced to the integer in the interval and the denoised label). We know that there exists a polynomial algorithm using Gaussian Elimination that finds a consistent hypothesis for this problem. We have already guaranteed that there is a point in $\mathcal{S}_m$ from every interval in the support of $\mathcal{D}_{S,\zeta}$. Further, $f$ is consistent on $\mathcal{S}_m$ and $f$ is constant in each of these intervals by design. Thus, with probability atleast $1 - \delta$ we have that $\mathcal{R}(f; \mathcal{D}_{S,\zeta}) = 0$.

By construction, $f$ makes a constant prediction on each interval $\left(j - \frac{1}{2}, j + \frac{1}{2}\right)$ for all $j \in \zeta$. Thus, for any perturbation radius $\gamma < \frac{1}{4}$ the adversarial risk $\mathcal{R}_{\mathrm{Adv},\mathcal{D}_{S,\prime\zeta}}(f) = 0$. Combining everything, we have shown that there is an algorithm that makes $2\,|\zeta|^2 \log \frac{2|\zeta|}{\delta}\left(8\frac{(1-\eta)}{(1-2\eta)^2} + 1\right)$ calls to the $\mathrm{EX}\left(\mathcal{D}_{S,\zeta}^\eta\right)$ oracle, runs in time polynomial in $|\zeta|, \frac{1}{1-2\eta}, \frac{1}{\delta}$ to return $f \in \mathcal{C}$ such that $\mathcal{R}(f; \mathcal{D}_{S,\zeta}) = 0$ and $\mathcal{R}_{\mathrm{Adv},\gamma}(f; \mathcal{D}_{S,\zeta}) = 0$ for $\gamma < \frac{1}{4}$. $\qquad\square$

**Lemma 4** (Union of Interval Concept Class). *There exists a learning algorithm $\widetilde{\mathcal{A}}$ such that given access to a noisy example oracle makes $m = O\left(|\zeta|^2 \frac{(1-\eta)}{(1-2\eta)^2} \log \frac{|\zeta|}{\delta}\right)$ calls to the oracle and returns a hypothesis $h \in \mathcal{H}$ such that training error is $0$ and with probability $1 - \delta$, $\mathcal{R}(f; \mathcal{D}_{S,\zeta}) = 0$.*

*Further for any $h \in \mathcal{H}$ that has zero training error on $m'$ samples drawn from $\mathrm{EX}^\eta(\mathcal{D}_{S,\zeta})$ for $m' > \frac{|\zeta|}{10\eta\gamma^2} \log \frac{|\zeta|}{10\gamma\delta}$ and $\eta \in \left(0, \frac{1}{2}\right)$ then $\mathcal{R}_{\mathrm{Adv},\gamma}(f; \mathcal{D}_{S,\zeta}) \geq 0.1$ for all $\gamma > 0$.*

*Proof of Lemma 4.* The first part of the algorithm works similarly to Lemma 3. The algorithm $\widetilde{\mathcal{A}}$ makes $m$ calls to the oracle $\mathrm{EX}(\mathcal{D}_s^m)$ to obtain a set of points $\mathcal{S}_m = \{(x_1, y_1), \cdots, (x_m, y_m)\}$ where $m \geq 2|\zeta|^2 \log \frac{2|\zeta|}{\delta}\left(8\frac{(1-\eta)}{(1-2\eta)^2} + 1\right)$. $\widetilde{\mathcal{A}}$ computes $h \in \mathcal{H}$ as follows. To begin, let the list of intervals in $h$ be $I$ and $\mathcal{M}_z = \{\}$ Then do the following for every $(x, y) \in \mathcal{S}_m$.

1. let $z := [x]$,

2. Let $\mathcal{N}_z \subseteq \mathcal{S}_m$ be the set of all $(x, y) \in \mathcal{S}_m$ such that $|x - z| < 0.5$.

3. Compute the majority label $\tilde{y}$ of $\mathcal{N}_z$.

4. Add all $(x, y) \in \mathcal{N}_z$ such that $y \neq \tilde{y}$ to $\mathcal{M}_z$

5. If $\tilde{y} = 1$, then add the interval $(z - 0.5, z + 0.5)$ to $I$.

6. Remove all elements of $\mathcal{N}_z$ from $\mathcal{S}_m$ i.e. $\mathcal{S}_m := \mathcal{S}_m \setminus \mathcal{N}_z$.

For reasons similar to Lemma 3, as $m \geq 2|\zeta|^2 \log \frac{2|\zeta|}{\delta}\left(8\frac{(1-\eta)}{(1-2\eta)^2} + 1\right)$, Lemma 5 guarantees that with probability $1 - \frac{\delta}{2}$, each interval will have atleast $\frac{8(1-\eta)}{(1-2\eta)^2} \log \frac{2|\zeta|}{\delta}$ samples. Then for any specific interval, Lemma 6 guarantees that with probability atleast $1 - \frac{2|\zeta|}{\delta}$, the majority vote for the label in that interval will succeed in returning the de-noised label. Applying a union bound over all intervals, will guarantee that with probability atleast $1 - \delta$, the majority label of every interval will be the denoised label. As each interval in $\zeta$ has atleast one point, all the intervals in $\zeta$ with label $1$ will be included in $I$ with probability $1 - \delta$. Thus, $\mathcal{R}(h; \mathcal{D}_{S,\zeta}) = 0$.

Now, for all $(x, y) \in \mathcal{M}_z$, add the interval $[x]$ to $I$ if $y = 1$. If $y = 0$ then $x$ must lie a interval $(a, b) \in I$. Replace that interval as follows $I := I \setminus (a, b) \cup \{(a, x), (x, b)\}$. As only a finite number of sets with Lebesgue measure of $0$ were added or deleted from $I$, the net test error of $h$ doesn't change and is still $0$ i.e. $\mathcal{R}(h; \mathcal{D}_{S,\zeta}) = 0$

For the second part, we will invoke Theorem 1. To avoid confusion in notation, we will use $\Gamma$ instead of $\zeta$ to refer to the sets in Theorem 1 and reserve $\zeta$ for the support of interval of $\mathcal{D}_{S,\zeta}$. Let $\Gamma$ be any set of disjoint intervals of width $\frac{\gamma}{2}$ such that $|\Gamma| = \frac{0.1|\zeta|}{\gamma}$. This is always possible as the total width of all intervals in $\Gamma$ is $\frac{0.1|\zeta|}{\gamma}\frac{\gamma}{2} = 0.1\frac{|\zeta|}{2}$ which is less than the total width of the support $\frac{|\zeta|}{2}$. $c_1, c_2$ from Eq. (3) is

$$c_1 = \mathbb{P}_{\mathcal{D}_{S,\zeta}}[\Gamma] = \frac{2 * 0.1|\zeta|}{2|\zeta|} = 0.1, \quad c_2 = \frac{2\gamma}{2|\zeta|}|\zeta| = \gamma$$

Thus, if $h$ has an error of zero on a set of $m'$ examples drawn from $\mathrm{EX}^\eta(\mathcal{D}_{S,\zeta})$ where $m' > \frac{0.1|\zeta|}{\eta\gamma^2} \log\left(\frac{0.1|\zeta|}{\gamma\delta}\right)$, then by Theorem 1, $\mathcal{R}_{\mathrm{Adv},\gamma}(h; \mathcal{D}_{S,\zeta}) > 0.1$.

Combining the two parts for

$$m > \max\left\{2|\zeta|^2 \log \frac{2|\zeta|}{\delta}\left(8\frac{(1-\eta)}{(1-2\eta)^2} + 1\right), \frac{0.1|\zeta|}{\eta\gamma^2} \log\left(\frac{0.1|\zeta|}{\gamma\delta}\right)\right\}$$

it is possible to obtain $h \in \mathcal{H}$ such that $h$ has zero training error, $\mathcal{R}(\mathcal{D}_{S,\zeta}; h) = 0$ and $\mathcal{R}_{\mathrm{Adv},\gamma}(h; \mathcal{D}_{S,\zeta}) > 0.1$ for any $\gamma > 0$.

$\square$

**Lemma 5.** *Given $k \in \mathbb{Z}_+$ and a distribution $\mathcal{D}_{S,\zeta}$, for any $\delta_2 > 0$ if $m > 2\left|\zeta\right|^2 k + 2\left|\zeta\right|^2 \log \frac{\left|\zeta\right|}{\delta_2}$ samples are drawn from $\mathrm{EX}\left(\mathcal{D}_{S,\zeta}\right)$ then with probability atleast $1 - \delta_2$ there are atleast $k$ samples in each interval $\left(j - \frac{1}{4}, j + \frac{1}{4}\right)$ for all $j \in \zeta$.*

*Proof of Lemma 5.* We will repeat the following procedure $\left|\zeta\right|$ times once for each interval in $\zeta$ and show that with probability $\frac{\delta}{\left|\zeta\right|}$ the $j^{th}$ run will result in atleast $k$ samples in the $j^{th}$ interval.

Corresponding to each interval in $\zeta$, we will sample atleast $m'$ samples where $m' = 2\left|\zeta\right| k + 2\left|\zeta\right| \log \frac{\left|\zeta\right|}{\delta_2}$. If $z_i^j$ is the random variable that is 1 when the $i^{th}$ sample belongs to the $j^{th}$ interval, then $j^{th}$ interval has atleast $k$ points out of the $m'$ points sampled for that interval with probability less than $\frac{\delta_2}{\left|\zeta\right|}$.

$$\mathbb{P}\left[\sum_i z_i^j \leq k\right] = \mathbb{P}\left[\sum_i z_i^j \leq (1 - \delta)\mu\right] \qquad \delta = 1 - \frac{k}{\mu}, \mu = \mathbb{E}\left[\sum_i z_i^j\right]$$

$$\leq \exp\left(-\left(1 - \frac{k}{\mu}\right)^2 \frac{\mu}{2}\right) \qquad \text{By Chernoff's inequality}$$

$$\leq \exp\left(-\left(\frac{m'}{2\left|\zeta\right|} - k + \frac{k^2 \left|\zeta\right|}{2m'}\right)\right) \qquad \mu = \frac{m'}{\left|\zeta\right|}$$

$$\leq \exp\left(k - \frac{m'}{2\left|\zeta\right|}\right) \leq \frac{\delta_2}{\left|\zeta\right|}$$

where the last step follows from $m' > 2\left|\zeta\right| k + 2\left|\zeta\right| \log \frac{\left|\zeta\right|}{\delta_2}$. With probability atleast $\frac{\delta}{\left|\zeta\right|}$, every interval will have atleast $k$ samples. Finally, an union bound over each interval gives the desired result. As we repeat the process for all $\left|\zeta\right|$ intervals, the total number of samples drawn will be atleast $\left|\zeta\right| m' = 2\left|\zeta\right|^2 k + 2\left|\zeta\right|^2 \log \frac{\left|\zeta\right|}{\delta_2}$. $\qquad \square$

**Lemma 6** (Majority Vote). *For a given $y \in \{0, 1\}$, let $S = \{s_1, \cdots, s_m\}$ be a set of size $m$ where each element is $y$ with probability $1 - \eta$ and $1 - y$ otherwise. If $m > \frac{8(1-\eta)}{(1-2\eta)^2} \log \frac{1}{\delta_1}$ then with probability atleast $1 - \delta_1$ the majority of $S$ is $y$.*

*Proof of Lemma 6.* Without loss of generality let $y = 1$. For the majority to be 1 we need to show that there are more than $\frac{m}{2}$ "1"s in $S$ i.e. we need to show that the following probability is less than $\delta_1$.

$$\mathbb{P}\left[\sum s_i < \frac{m_1}{2}\right] = \mathbb{P}\left[\sum s_i < \frac{m_1}{2\mu} * \mu + \mu - \mu\right] \qquad \mu = \mathbb{E}\left[\sum s_i\right]$$

$$= \mathbb{P}\left[\sum s_i < \left(1 - \left(1 - \frac{m_1}{2\mu}\right)\right)\mu\right]$$

$$\leq \exp\left(-\frac{(1 - 2\eta)^2}{8(1-\eta)^2}\mu\right) \qquad \text{By Chernoff's Inequality}$$

$$= \exp\left(-\frac{(1 - 2\eta)^2}{8(1-\eta)}m\right) \qquad \because \mu = (1 - \eta)m$$

$$\leq \delta_1 \qquad \because m > \frac{8(1-\eta)}{(1-2\eta)^2}\log\frac{1}{\delta_1}$$

$\qquad \square$

## C  ADDITIONAL EXPERIMENTS

### C.1  ROBUST TRAINING IGNORES RARER SUB-POPULATIONS (SYNTHETIC SETTING)

This phenomenon is demonstrated more clearly in a simpler distribution for different NN configurations in Figure 7. We create a binary classification problem on $\mathbb{R}^2$. The data is uniformly supported

| $\epsilon$ | Train-Acc. (%) | Test-Acc (%) |
|------------|----------------|--------------|
| 0.0 | 99.98 | 95.25 |
| 0.25 | 97.23 | 92.77 |
| 1.0 | 86.03 | 81.62 |

Table 1: Train and test accuracies on clean CIFAR10 for ResNet-50 trained using $\ell_2$ $\epsilon$-adversaries. The $\epsilon = 0$ setting represents the natural training.

on non-overlapping circles of varying radiuses. All points in one circle have the same label i.e. it is either blue or red depending on the color of the circle. We train a shallow network with 2 layers and 1000 neurons in each layer (Shallow-Wide NN) and a deep network with 4 layers and 100 neurons in each layer using cross entropy loss and SGD. The background color shows the decision region of the learnt neural network. Figure 7 shows that the adversarially trained (AT) models ignore the smaller circles (i.e. rare sub-populations) and tries to get a larger margin around the circles it does classify correctly whereas the naturally trained (NAT) models correctly predicts every circle but ends up with very small margin around a lot of circles.

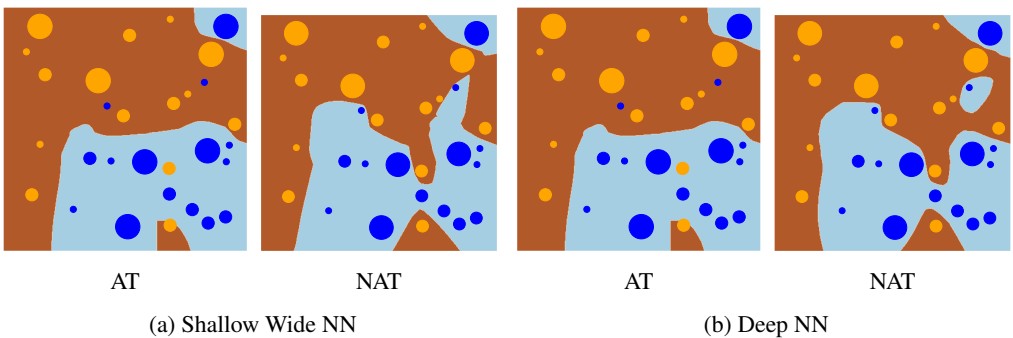

|         AT         |         NAT        |         AT         |         NAT        |

|     (a) Shallow Wide NN     |     (b) Deep NN     |

Figure 7: Adversarial training (AT) leads to larger margin, and thus adversarial robustness around high density regions (larger circles) but causes training error on low density sub-populations (smaller circles) whereas naturally trained models (NAT) minimizes the training error but leads to regions with very small margins.

## C.2 COMPLEXITY OF DECISION BOUNDARIES

When neural networks are trained they create classifiers whose decisions boundaries are much simpler than they need to be for being adversarially robust. A few recent papers (Nakkiran, 2019; Schmidt et al., 2018) have discussed that robustness might require more complex classifiers. In Theorem 2 and 3 we discussed this theoretically and also why this might not violate the traditional wisdom of Occam's Razor. In particular, complex decision boundaries does not necessarily mean more complex classifiers in statistical notions of complexity like VC dimension. In this section, we show through a simple experiment how the decision boundaries of neural networks are not "complex" enough to provide large enough margins and are thus adversarially much more vulnerable than is possible.

We train three different neural networks with ReLU activations, a shallow network (Shallow NN) with 2 layers and 100 neurons in each layer, a shallow network with 2 layers and 1000 neurons in each layer (Shallow-Wide NN), and a deep network with 4 layers and 100 neurons in each layer. We train them for 200 epochs on a binary classification problem as constructed in Figure 8. The distribution is supported on blobs and the color of each blob represent its label. On the right side, we have the decision boundary of a large margin classifier, which is simulated using a 1-nearest neighbour.

From Figure 8, it is evident that the decision boundaries of neural networks trained with standard optimizers have far *simpler* decision boundaries than is needed to be robust (eg. the 1- nearest neighbour is much more robust than the neural networks.)

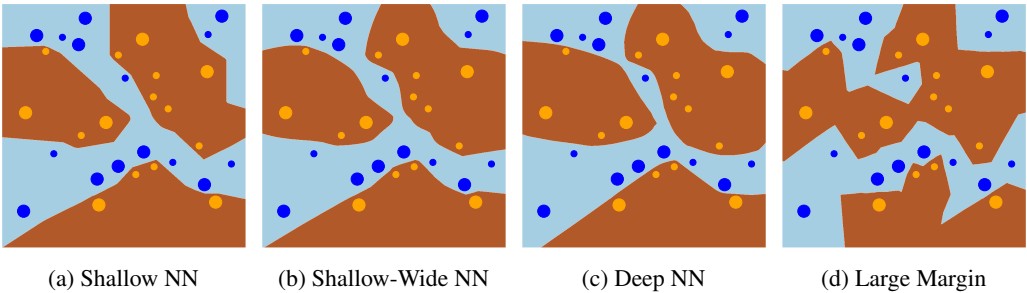

| (a) Shallow NN | (b) Shallow-Wide NN | (c) Deep NN | (d) Large Margin |

Figure 8: Decision boundaries of neural networks are much simpler than they should be.

### C.3 ACCOUNTING FOR FINE GRAINED SUB-POPULATIONS LEADS TO BETTER ROBUSTNESS

One way to evaluate whether more meaningful representations lead to better robust accuracy is to use training data with more fine-grained labels (e.g. subclasses of a class); for example, one would expect that if different breeds of dogs are labelled differently the network will learn features that are relevant to that extra information. We show both using synthetic data, CIFAR100 (Krizhevsky & Hinton, 2009), and Restricted Imagenet (Tsipras et al., 2019) that training on fine-grained labels does increase robust accuracy.

We hypothesize that learning more meaningful representations by accounting for fine-grained sub-populations within each class may lead to better robustness. We use the theoretical setup presented in Appendix A and Figure 6b. However, if each of the circles belonged to a separate class then the decision boundary would have to be necessarily more complex as it needs to, now, separate the balls that were previously within the same class. We test this hypothesis with two experiments. First, we test it on the the distribution defined in Theorem 3 where for each ball with label 1, we assign it a different label (say $\alpha_1, \cdots, \alpha_k$) and similarly for balls with label 0, we assign it a different label ($\beta_1, \cdots, \beta_k$). Now, we solve a multi-class classification problem for $2k$ classes with a deep neural network and then later aggregate the results by reporting all $\alpha_i$s as 1 and all $\beta_i$s as 0.The resulting decision boundary is drawn in Figure 9a along with the decision boundary for natural training and AT. Clearly, the decision boundary for AT is the most complex and has the highest margin (and robustness) followed by the multi-class model and then the naturally trained model.

Second, we also repeat the experiment with CIFAR-100. We train a ResNet50 (He et al., 2016) on the fine labels of CIFAR100 and then aggregate the fine labels corresponding to a coarse label by summing up the logits. We call this model the *Fine2Coarse* model and compare the adversarial risk of this network to a ResNet-50 trained directly on the coarse labels. Note that the model is end-to-end differentiable as the only addition is a layer to aggregate the logits corresponding to the fine classes pertaining to each coarse class. Thus PGD adversarial attacks can be applied out of the box. Figure 9b shows that for all perturbation budgets, *Fine2Coarse* has smaller adversarial risk than the naturally trained model.

We also repeat the experiment with Restricted Imagenet (Tsipras et al., 2019) where we obtain the fine and coarse class as mentioned in Table 2. There are 60 fine classes and 10 coarse classes with each coarse class having 6 distinct fine classes in them. The train set size is 77237 and the test set size is 3000. The fine classes within each coarse are balanced i.e. given a coarse class all the fine classes in it are equally represented in this dataset.

## D EXPERIMENTAL DETAILS

In this section, we will discuss details about the experiments and model architectures used to help make sure that the experiments are reproducible.

### D.1 MODEL ARCHITECTURES IN SECTION 3.1

The model architecture used for MNIST experiments has four convolutional layers, followed by two fully connected layers. The first four convolutional layers have $32, 64, 128, 256$ output filters

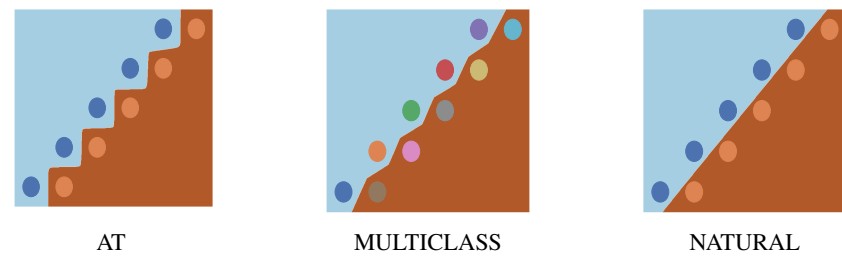

|     |     |     |
| :---: | :---: | :---: |
| AT | MULTICLASS | NATURAL |

(a) Decision Region of neural networks are more complex for adversarially trained models. Treating it as a multi-class classification problem, with natural training (MULTICLASS), also increases robustness by increasing the margin.

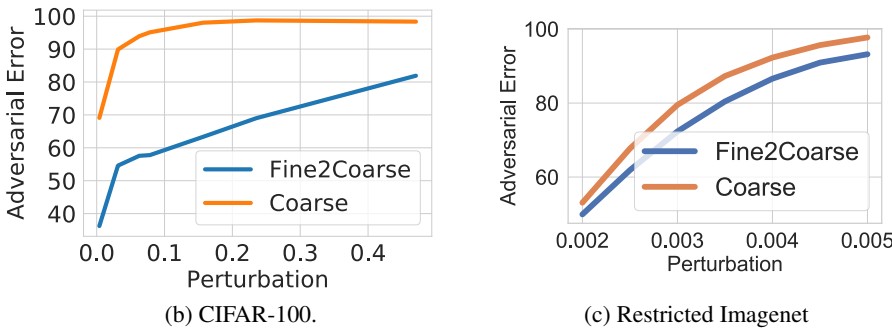

| (b) CIFAR-100. | (c) Restricted Imagenet |
| :---: | :---: |

Figure 9: Assigning a separate class to each sub-population within the original class during training increases robustness by learning more meaningful representations.

| Coarse Class | Fine Classes |
| :---: | :--- |
| Dog | Chihuahua, Japanese spaniel, Maltese dog, Pekinese, Shih-Tzu, Blenheim spaniel |
| Bird | cock, hen, ostrich, brambling, goldfinch, house finch |
| Insect | tiger beetle,ladybug,ground beetle, long-horned beetle, leaf beetle, dung beetle |
| Monkey | guenon, patas, baboon, macaque, langur, colobus |
| Car | jeep, limousine,cab, beach wagon, ambulance, convertible |
| Feline | leopard, snow leopard, jaguar, lion, cougar, lynx |
| Truck | tow truck, moving van, fire engine, pickup, garbage truck, police van |
| Fruit | Granny Smith, rapeseed, corn, acorn, hip, buckeye |
| Fungus | agaric, gyromitra, stinkhorn, earthstar, hen-of-the-woods, coral fungus |
| Boat | gondola, fireboat, speedboat, lifeboat, yawl, canoe |

Table 2: Fine-grained classes in Restricted Imagenet

and $3, 4, 3, 3$ sized kernels respectively. This is followed by a fully connected layers with a hidden dimension of $1024$. The network is optimized with SGD with a batch size of $128$, learning rate of $0.1$ for $60$ epochs and learning rate is decreased to $0.01$ after $50$epochs.

For experiments on CIFAR we use standard VGG19, ResNet18 and DenseNet architectures and training procedures.

### D.2 DETAILS ON EXPERIMENTS ON MEMORIZATION AND INFLUENCE

For a given dataset $\mathcal{S}$, training algorithm $\mathcal{A}$, Zhang & Feldman (2020) measure the label memorization by $\mathcal{A}$ on $S$ using two related quantities of memorization and influence. For a training example $(x_i, y_i) \in \mathcal{S}$ and a test example $(x'_j, y'_j)$, the two quantities are a special case of measuring the influence of $(x_i, y_i)$ on the expected accuracy at some example $z = (x, y)$. Influence on test example

measures the impact of $(x'_j, y'_j)$ on the expected accuracy of $(x'_j, y'_j)$. Memorization corresponds to the influence of example $(x_i, y_i)$ on the accuracy on itself (or self-influence).

**Memorization or Self-Influence:** Self influence of an example with respect to an algorithm (model, optimizer etc) can be defined as how unlikely it is for the model learnt by that algorithm to be correct on an example if it *had not seen* that example during training compared to if it *had seen* the example during training. It can be formalized as followed which is borrowed from Eq (1) in Zhang & Feldman (2020)

Memorization by $\mathcal{A}$ on example $(x_i, y_i) \in \mathcal{S}$ is defined as

$$\text{mem}(\mathcal{A}, \mathcal{S}, i) := \mathbb{P}_{h \leftarrow \mathcal{A}(\mathcal{S})}[h(x_i) = y_i] - \mathbb{P}_{h \leftarrow \mathcal{A}(\mathcal{S}^{\backslash i})}[h(x_i) = y_i]$$

where $\mathcal{S}^{\backslash i}$ denotes the dataset $\mathcal{S}$ with $(x_i, y_i)$ removed, $h \leftarrow \mathcal{A}(\mathcal{S})$ denotes the model $h$ obtained by training using algorithm $\mathcal{A}$ (which includes the model architecture) on the dataset $\mathcal{S}$ and the probability is taken over the randomization inherent in the training algorithm $\mathcal{A}$.

**Influence of a training sample:** Intuitively, it measures the probability that a certain test example would be classified correctly if the model were learned using a training set that *did not contain* the training point compared to if the training set *did contain* that particular training point. This can be defined as follows which is borrowed from Eq 2 in Zhang & Feldman (2020). Using a similar notation as memorization the influence of $(x_i, y_i)$ on $(x'_j, y'_j)$ can be measured as

$$\text{infl}(\mathcal{A}, \mathcal{S}, (x_i, y_i), (x'_j, y'_j)) := \mathbb{P}_{h \leftarrow \mathcal{A}(\mathcal{S})}[h(x'_j) = y'_j] - \mathbb{P}_{h \leftarrow \mathcal{A}(\mathcal{S}^{\backslash i})}[h(x'_j) = y'_j]$$

We found the images in Figure 4 by manually searching for each test image, the training image that is misclassified and is visually close to it. Our search space was shortened with the help of the influence scores each training image has on the classification of a test image. We searched in the set of top-10 most influential mis-classified train images for each mis-classified test image. The model used for Figure 4 is a AT model for CIFAR10 with $\ell_2$-adversary with an $\epsilon = 0.25$ and a model trained with TRADES for MNIST with $\lambda = \frac{1}{6}$ and $\epsilon = 0.3$. Zhang & Feldman (2020) had provided us the with the memorization scores for each image in CIFAR10 as well as the influence score of each training image on each test image for each class in CIFAR-10. High Influence pairs of Imagenet were obtained from `https://pluskid.github.io/influence-memorization/`. This was used to obtain the figures for Imagenet in Figure 4.

