# OpenReview forum: "How Benign is Benign Overfitting ?"
_ICLR.cc/2021/Conference — ICLR 2021 Spotlight_

### Official Review · AnonReviewer1 · 2020-10-19
**Excellent motivation with useful empirical results but the main results might not be interesting**

**Rating:** 6
**Confidence:** 3

**Review:**

The main contribution of the paper is to study the connection between adversarial robustness, on the one hand, and label noise & data representation on the other hand. Here, an algorithm is said to be robust if for every training example xi with label yi, one cannot find an instance x within a small distance of xi that is assigned a different label from yi by the model. This is a standard definition of robustness in the literature.

The authors claim that one primary source of the lack of adversarial robustness is label noise. They show this via a simple construction. Suppose you have a sphere of instances of radius r that have the label y (true label). Suppose then that you flip some of the labels at random. Then, obviously, any algorithm that has a perfect training accuracy will not be robust because, by construction, there exists examples within a distance of 2r from each other that have different labels. The authors make this argument formal. This result by itself is not very interesting. What would be interesting is if label noise was indeed the main source of lack of robustness in models trained on standard datasets, e.g. CIFAR10 or ImageNet. The authors show that such datasets do contain a lot of label noise but, unfortunately, state that removing such label noise by itself is not sufficient to make the trained model robust. In fact, some of their results indicate that robustness is not caused by label noise. For example, all ResNet18, DenseNet121, and VGG19 trained on CIFAR10 have a large adversarial error even with zero label noise! (Figure 2 bottom)

Then, the authors discuss a different argument: they show that adversarial training prevents memorization of mislabeled examples. This is interesting. However, it would be great if the author could clarify how they determined that 994 out of the 997 samples in CIFAR10 mentioned in Section 3.2 were mislabelled. Did they manually inspect all 994 examples? This would correspond to about 2% of CIFAR10 training examples. I find this number to be quite large. For instance, the BiT model (https://arxiv.org/pdf/1912.11370.pdf) achieves more than 99% test accuracy on CIFAR10. If you look into their analysis of errors in Fig 8, it suggests that the ratio of mislabeled examples in CIFAR10 is less than 0.5%, not 2%. I would appreciate it if the authors could clarify how they determined that 994 out of 997 were mislabelled.

The second main contribution of the paper is on the relation between adversarial training and representation. The main conclusion here is that if one chooses the hypothesis space carefully, the learner can achieve good generalization and robustness. But, if the hypothesis space is not chosen well, one can achieve good generalization but poor robustness. So, the choice of the hypothesis space matters. I find this result to be disappointing. Here is a much simpler example. Suppose we have two spheres that are well-separated from each other, each sphere corresponds to a class, and we add some small label noise. Now, consider the class of linear separators, e.g. using large-margin SVM. These will not fit the training data perfectly but will have a small excess Bayes risk (since the Bayes-optimal decision boundary are disjoint spheres) and are robust. On the other hand, let the second hypothesis space be the space induced by the kNN classifier. Both the training and test error will be small (since kNN averages the labels of neighbours and the label noise is small), but the adversarial error is large (with a high probability, there will be a small region in each sphere that is predicted differently by kNN).

In summary, I think the paper has an excellent motivation with useful empirical results (e.g. Figure 2), but the main results might not be interesting since one can arrive at such conclusions using much simpler arguments as I mentioned above.

Additional Remarks:
- The authors suggest in the Introduction section that there is no fundamental tradeoff between robustness and accuracy. Later in Section 3.2, however, they point out that robust training ignores rare examples, which reduces the test accuracy. They argue for this using the notion of “self-influence”. I suggest revisiting that paragraph in the introduction.

- The authors add a claim in the Conclusion section that is not discussed within the main text as far as I could tell. They state that some invariances can increase adversarial vulnerability. Where is this mentioned in the main text?

- There are a few typos in the paper:
	* Abstract: “partsub-“optimal —> part sub-optimal
	* Page 4: “Thus, smaller the value of” —> “Thus, the smaller the value of”.
	* Page 6: “we found that that” —> “we found that”
	* Page 7: “that are are heavily” —> “that are heavily”

---

> ### Author Response · Authors · 2020-11-22
> **Response to Reviewer 1**
>
> * **"How they determined that 994 out of the 997 samples…"** - We believe there might be some misunderstanding. The experiment is on MNIST and the samples were mis-labelled by us. This is mentioned in the same paragraph -- “We also observe this in a synthetic experiment on the full MNIST dataset where we assigned random labels to 15% of the dataset.”.
> * **Regarding the result in Theorem 1**, we think, and the other reviewers have also agreed, that the phenomenon is relevant and it is important to not have to rely on intuitions but instead have proper mathematical and experimental evidence, which is what we tried to provide here.
> * **Alternate proof of using Theorem 2 using KNN** We thank the reviewer for providing an interesting alternative proof idea for Theorem 2 using K-nearest neighbour as the non-robust classifier. However, proving such a result formally would also require more careful arguments and specific relation between k and the ambient dimension.
>
>    For good generalization, the gap in sample complexity between KNN  and the linear classifier is significantly greater than the example in Theorem 2.
>
>     Secondly, having low test error for KNN will require k to be large enough (For example if  k=1, points for which mis-labelled data is the nearest neighbour will be mis-classified.) However for large k, KNN will also be a robust classifier. Thus, it seems to us that either KNN will have high test error and high adversarial error or low test error and low adversarial error whereas the example in Theorem 2 has high adversarial error but low test error.
>
>     It might be possible  to refine the idea into a proof that mirrors the statement of Theorem 2 but will possibly end up being as complicated as the proof of Theorem 2.
>
>
> We have incorporated the rest of your minor suggestions in the draft. Thank you very much for reading the paper in detail and the suggestions. They are thought-provoking and we will think about adapting this example in detail.

---

### Official Review · AnonReviewer4 · 2020-10-22
**Nice attempt of exploring both theoretically and empirically the causes of lack of adversarial robustness of benign overfitting**

**Rating:** 7
**Confidence:** 3

**Review:**


The goal of the paper is to investigate both theoretically and empirically the reasons of vulnerability of overparameterized classifiers obtained by the so called “benign overfitting”. More precisely, two causes of adversarial vulnerability are underlined: label noise memorization and sub-optimal representation learning. The first theorem of the paper shows that for some data generating distributions, even a small fraction of label noise leads to an adversarial prediction risk bounded away from zero for any classifier having zero training error and for any sufficiently large sample size. The second theorem shows that in the presence of label noise the choice of the overparameterized family (the representation) is very important. Namely, while for a good representation one may have “training error = test error = adversarial error = 0”, for another representation it holds that “training error = test error = 0” but “adversarial error > 0.1”. This theoretical results are illustrated by extensive experimental results.

I find the paper very well written. In my opinion, it will be of interest for most participants of ICLR. It is of course not surprising that label noise memorization and poor representation learning cause adversarial vulnerability, but the way it is theoretically quantified and empirically demonstrated in this paper is worth being published.

Minor remarks
Abstract : “in partsub-optimal” -> “in part sub-optimal”
Line 2 of Thm 1: D in the subscript of P should be \mathcal D.
Proof of Thm 1: in the lines below (4), “P_{S_m ∼D^m}” should be removed (4 occurrences)
Proof of Thm 1: in the chain of equalities/inequalities below eq (4), the fourth line should be an inequality.

---

> ### Author Response · Authors · 2020-11-22
> **Response to Reviewer 4**
>
> Thank you for your comments. We have included most of the minor suggestions that you made.
>
> In the proof of Theorem 1, we think the fourth line should indeed be an equality as $P(x\in\bigcup (S_1, S_2,..., S_k))  = P (\exists i \in (1,2,....,k ) ~ \mathrm{s.t.}~ x \in S_k)$.
>
> We have also kept the $P_{S_m \sim D^m}$ as we wanted to emphasize that the  $1 - \delta$ probability bound is with respect to the random draw and mislabelling of $S_m$.

---

> > ### Comment · AnonReviewer4 · 2020-11-25
> > **inequality in the proof of Theorem 1**
> >
> > I still believe that it should be an inequality. While I agree that $x\in \cup_{i=1}^k S_i$ is equivalent to $\exists i\in\{1,\ldots,k\}$ such that $x\in S_i$, I have to draw your attention to the fact that in your formulae there is also a second event that changes from line 3 to line 4. If you know that  $z\in\mathcal B_\rho(x)$ and $x\in\mathcal B_\rho(s)$, this implies that  $z\in\mathcal B_{2\rho}(x)$ (by the triangle inequality). The converse is generally not true.

---

> > > ### Author Response · Authors · 2020-11-25
> > > **Thanks for noting**
> > >
> > > Thank you for noting that. You are right and we will change it in the next revision of the paper and write a line below the proof to explain it.

---

### Official Review · AnonReviewer3 · 2020-10-27

**Rating:** 7
**Confidence:** 3

**Review:**

The main contribution of this paper as I see it is in pointing out that label noise can negatively affect the adversarial robustness of interpolating predictors, even when the standard 0-1 error is small. The paper supports this claim with a simple theoretical construction (Theorem 1) and several empirical experiments. The paper also argues that adversarial training techniques avoid memorizing noisy labeled examples and  rare examples which partly explains why adversarial training incurs higher standard 0-1 error.

Another result of this paper is Theorem 2, which exhibits an example of a learning problem and two function classes $C$ and $H$, where: (1)  there is a classifier in $C$ that interpolates the training data and furthermore achieves zero standard error and zero robust error, and (2) there is a classifier in $H$ that interpolates the training data and achieves zero standard error but has high robust error. The result would be stronger  if the quantifier in (1) is strengthened to: for any classifier in $C$ that interpolates the training data (rather than there exists). Furthermore, as the authors mention in related work, (Montasser, Hanneke, Srebro, 2019) have shown that there are function classes that are robustly learnable but only improperly. So, it is kind of already known that the representations used for learning matter for adversarial robustness. It would be good the authors could explain the difference between their contribution and what’s known before.

Some questions:
It would be interesting to see if adversarial training can be made such that to achieve zero robust loss on the training data, which means that it interpolates the training data. What would be the standard 0-1 error of such predictors?
Would it be possible to strengthen Theorem 1 by relaxing the condition in Equation (3) such that its only required that the mass of $\zeta$ under $D$ is at least $c_1$ (rather than requiring union of the perturbation balls to have mass at least $c_1$)?

---

> ### Author Response · Authors · 2020-11-22
> **Response to Reviewer 3**
>
> We thank the reviewer for their reading of the paper and the helpful comments.
>
> *  **Classifiers in C do not interpolate** - In the setting of Theorem 2, classifiers in $\mathcal{C}$ in fact do not interpolate the training data but they minimize the training loss. Classifiers in $\mathcal{H}$ can interpolate the training data and there, indeed, we have the “For all $h$ in $\mathcal{H}$ that gets zero train error, adversarial error will be large”
> * **"For all" statement in Theorem 2** Due to the structure of the label noise, one can appeal to standard VC bounds to show that the minimizer in $\mathcal{C}$ for this setting is unique with a high probability and then the result can indeed be extended to “For all c in C…”
> *  **Regarding the conditions in Theorem 1**  Requiring the mass of $\zeta$ under $\mathcal{D}$ to be at least $c_1$ is actually a stronger condition as it would mean that  either $\zeta$ is an uncountably infinite set or there are point masses in the distribution
>    * If the intended meaning was that each of the perturbation balls around points in $\zeta$ have a mass of $\frac{c_1}{|\zeta|}$ i.e. just keep the second assumption and ignore the first condition. That wouldn't work as all the points in $\zeta$ might lie very close to each other and, in effect all of them would affect the same region in $\mathcal{D}$.
>    * The other interpretation would be that each of the perturbation balls around points in zeta have a mass of $c_1$- this is also a stronger assumption as a) it automatically satisfies the union assumption b) the  RHS of the second constraint is (3) would be much bigger here i.e. $|\zeta|$ times the current condition.
> * **Discussion of Montasser et. al. 2019** - Thank you for mentioning that. We have included a slightly longer discussion in the Related Works section in the appendix to highlight the differences but in short, the improper learning results in Montasser et. al. 2019 requires larger sample complexities whereas in our case learning algorithms for both $\mathcal{C}$ and $\mathcal{H}$ have similar sample complexities (in Theorem 3, the class $\mathcal{C}$, learning which yields a robust classifier, actually has a smaller VC dimension than $\mathcal{H}$). The second difference is the presence of label noise in the training dataset in Theorem 2.

---

### Official Review · AnonReviewer2 · 2020-10-28
**Easy to read paper clearly demonstrating some intuitive insights on generalization and adversarial robustness**

**Rating:** 8
**Confidence:** 4

**Review:**

### Summary

The generalization ability of networks with zero training error has been heavily studied.  This paper extends beyond generalization to test sets to study the network's robustness to adversarial examples.  The paper provides two theoretical contributions demonstrating that a very low training error can indicate poor robustness under reasonable conditions.  They illustrate this with experiments using label noise, demonstrating that adversarially robust networks spurn overfitting on incorrectly labelled data.  They additionally experimentally demonstrate that unusual training examples, even if correctly labelled, are unlikely to be correctly predicted by adversarially robust networks.

### Significance

The generalization properties of neural networks and adversarial robustness are two very fast-moving areas of machine learning.  This paper does a nice job revealing some properties of overfit networks.  These properties are intuitive (at least, I would have assumed them), but I have not seen them so nicely laid out, and it is important to not have to assume.  It does a great job of filling in these holes with evidence, and so I find it quite significant.

### Originality

To my knowledge, the work is original.

### Quality

The experiments are quite well designed and performed.  I find the second theoretical contribution too quickly discussed, and the "unusual examples" experiment insufficiently emphasized, but otherwise it is quite a good paper.  Graphs and figures are meaningful and well explained.  Theoretical results nicely support portions of the paper that might otherwise be criticized as anecdotal.

### Clarity

Very clearly written.

---

> ### Author Response · Authors · 2020-11-22
> **Response to Reviewer 2**
>
> We thank the reviewer for appreciating the originality and clarity of our work. Due to the limited space, we had to choose which part we wanted to highlight more and we chose the label noise. If we are allowed more space in the camera-ready version, we will expand more on the representation learning part.

---

### Author Response · Authors · 2020-11-22
**General Comment**

We thank all the reviewers for their encouraging comments and reviews and we are glad that you found the paper interesting, well-written and relevant. We have made some changes to the paper as requested by the reviewers to correct the typos and include a small discussion to highlight the differences with the examples from Montasser et. al. 2019.

We will respond to individual questions from the reviewers separately below.

---

### Comment · ~Cheng-Han_Chiang1 · 2021-02-08
**Questions Regarding the Proof of Theorem 1**

Thank you for the wonderful paper!
When I am going through the proof of Thm 1, I have some questions and I will be appreciate if the authors can help me with them.

In the first line of p.17, it is written that $\mathcal{R}_{Adv,2\rho}(f;\mathcal{D})=\mathbb{P} _{\mathcal{S_m \sim D^m}}[\mathbb{P} _{{(\textbf{x}, y)\sim \mathcal{D}}}[ \exists \textbf{z}\in\mathcal{B} _{2\rho}(\textbf{x}) \wedge y\neq f(\textbf{z})]]$.
I don't quite understand why we need to take the probability $\mathbb{P} _{\mathcal{S_m \sim D^m}}$, since this is not presented in the Def 1. As far as I understand, the adversarial error is defined on a fixed $f$ (a fixed classifier), but taking the probability over $\mathcal{S_m \sim D^m}$ indicates that we are considering the classifiers trained using different $\mathcal{S_m}$. Am I misunderstanding something?

And I am not getting why the second equality holds, i.e. $\mathbb{P} _{\mathcal{S_m \sim D^m}}[\mathbb{P} _{{(\textbf{x}, y)\sim \mathcal{D}}}[ \exists \textbf{z}\in\mathcal{B} _{2\rho}(\textbf{x}) \wedge y\neq f(\textbf{z})]]=\mathbb{P} _{\mathcal{S_m \sim D^m}}[\mathbb{P} _{{(\textbf{x}, y)\sim \mathcal{D}}}[ \exists \textbf{z}\in\mathcal{B} _{2\rho}(\textbf{x}) \wedge c(\textbf{z})\neq f(\textbf{z})]]$. I can understand the explanation below that says "The second equality follows from the assumptions that each of the balls around  $s\in\zeta$ are pure in their labels"; however, the previous statement only holds if the $\textbf{x}$ is sampled from $\mathcal{D}$ falls in the $\mathcal{B} _{\rho} ^p (\textbf{s})$ ball. Can we guarantee that the $(\textbf{x}, y)$ sampled from $\mathcal{D}$ always lies in the $\mathcal{B} _{\rho} ^p (\textbf{s})$ ball?

---

> ### Author Response · Authors · 2021-02-08
> **Thanks and clarifications**
>
> Dear Cheng-Han,
>
> We are glad you liked our paper.
>
> Regarding your first query for why $\mathcal{S}_m\sim\mathcal{D}^m$ is necessary, this is necessary to provide a high-probability guarantee of $1-\delta$. For example, it is possible that with a very small probability $\delta$, a dataset $\mathcal{S}_m$ is sampled such that $f$ when overfit on that dataset does not suffer the mentioned level of adversarial error. One example of such an $\mathcal{S}_m$ would be where all the mis-labelled points in the training dataset lie within a small $\rho$ ball thereby only causing adversarial mis-classification for points in and around that ball. However, the result shows that with a high probability over the sampling of the dataset, any classifier $f$ ($f$ is dependent on the dataset) that overfits to that sampled dataset the adversarial error will be high.
>
> For your second point, I believe you are referring to what has been pointed to by Reviewer 4 below. It should indeed be an inequality. We will fix it in the final version.

---

> > ### Comment · ~Cheng-Han_Chiang1 · 2021-02-13
> > **Thanks and further a question**
> >
> > Thank you for your fast reply and clear clarification.
> > The clarification you gave perfectly answer my question 1.
> >
> >
> > As for my second question, I think reviewer 4 is referring to the 4th line in the chain of the equalities, but my question is about the 2nd equality in the chain of equalities.
> > If what reviewer 4 pointed out also includes that the 2nd equality should not hold, then I think I am referring to the same thing as reviewer 4.
> > Thanks again for your reply and this wonderful paper.

---

### Decision · Program_Chairs · 2021-01-07
**Final Decision**

**Decision:**

Accept (Spotlight)

**Comment:**

The paper seeks to understand how training over-parametrized models (e.g., those based on neural networks) to zero training accuracy even when the test error is small (i.e., benign overfitting) can introduce vulnerabilities in the form of adversarial examples and how to remedy the situation. The paper implicates label noise as one of the causes of adversarial robustness, and suboptimal representations learned as part of the training as another. The claims are supported both theoretically and empirically. A good paper overall, accept!